# Finetuning CLIP to Reason about Pairwise Differences

**Dylan Sam**    *dylansam@andrew.cmu.edu*
*Carnegie Mellon University*

**Devin Willmott**    *devin.willmott@us.bosch.com*
*Bosch Center for AI*

**Joao D. Semedo**    *joao.semedo@us.bosch.com*
*Bosch Center for AI*

**J. Zico Kolter**    *zkolter@cs.cmu.edu*
*Carnegie Mellon University*

**Reviewed on OpenReview:** *https://openreview.net/forum?id=USNJFZTWPn*

## Abstract

Vision-language models (VLMs) such as CLIP are trained via contrastive learning between text and image pairs, resulting in aligned image and text embeddings that are useful for many downstream tasks. A notable drawback of CLIP, however, is that the resulting embedding space seems to lack some of the structure of their purely text-based alternatives. For instance, while text embeddings have been long noted to satisfy *analogies* in embedding space using vector arithmetic, CLIP has no such property. In this paper, we propose an approach to natively train CLIP in a contrastive manner to reason about differences in embedding space. We finetune CLIP so that *text descriptions of differences between images* correspond to their difference in image embedding space, using synthetically generated data with large language models on image-caption paired datasets. We first demonstrate that our approach yields significantly improved capabilities in ranking images by a certain attribute (e.g., elephants are larger than cats), which is useful in retrieval or constructing attribute-based classifiers, and improved zeroshot classification performance on many downstream image classification tasks. In addition, our approach enables a new mechanism for inference that we refer to as comparative prompting, where we leverage prior knowledge of text descriptions of differences between classes of interest, achieving even larger performance gains in classification. Finally, we illustrate that the resulting embeddings obey a larger degree of geometric properties in embedding space, such as in text-to-image generation.

## 1 Introduction

Vision-language models (VLMs) (Jia et al., 2021; Li et al., 2022a), and more specifically CLIP (Radford et al., 2021), leverage paired instances of images and corresponding text descriptions to produce a general-purpose joint embedding between images and language. These models have created a new paradigm of prompting (Radford et al., 2021; Li & Liang, 2021; Bach et al., 2022). In this new paradigm, we can easily design image classifiers through text descriptions of classes and by selecting which of our class descriptions most closely aligns with an image (in terms of cosine similarity in the multimodal embedding space). These models, when paired with additional training and a diffusion model, can also generate images corresponding to user-specified text prompts (Podell et al., 2023). Ultimately, this paradigm *fundamentally* relies on the accurate alignment of image and text modalities.

While contrastive-based pretraining on large datasets aims to achieve this embedding alignment, a notable drawback of CLIP models is that they do not exhibit the structure of purely language-based embeddings. For

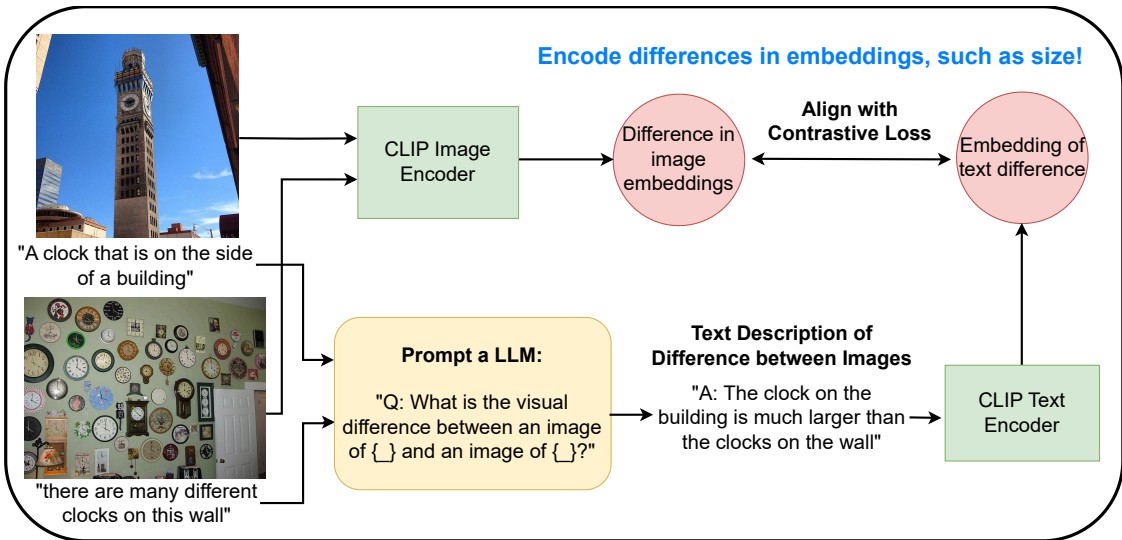

Figure 1: Our approach (**PC-CLIP**) to improve CLIP's ability to reason about differences. We use LLMs to describe the visual difference between a pair of captions, and align the difference in CLIP's image embeddings with a text embedding of this synthetic difference via a contrastive loss.

instance, text embeddings satisfy *analogies* in embedding space using vector arithmetic, e.g., Text("*King*") - Text("*Man*") + Text("*Woman*") ≈ Text ("*Queen*") (Mikolov et al., 2013), while CLIP has no such property. In addition to these shortcomings, previous works demonstrate that CLIP's embeddings lack geometric properties (Goel et al., 2022), exhibit large gaps between different modalities (Liang et al., 2022), and struggle with handling more complex descriptions, such as connections between multiple attributes and objects (Lewis et al., 2022). As CLIP is commonly used as a backbone for a wide variety of tasks (Ramesh et al., 2022; Podell et al., 2023; Bain et al., 2022), accurately encoding meaningful differences between images can lead to benefits in many downstream tasks, such as compositional text-to-image generation or retrieval.

In this paper, we propose to align the difference between CLIP's text encodings of meaningful text descriptions of differences with the corresponding differences in image embeddings, to improve its ability to reason about differences. We show that these text differences between images are poorly localized in CLIP's embedding space (see our experiments in Section 4.2). On the contrary, prior work has shown that large language models (LLMs) can generate meaningful differences between concepts (Howard et al., 2023). We thus use LLMs to generate a synthetic dataset of text descriptions of the differences between pairs of images from an image-caption paired dataset (e.g., COCO (Lin et al., 2014)). We then finetune CLIP to align these comparisons with the differences in CLIP's frozen image embeddings via a contrastive objective. This process, which we refer to as **PC-CLIP** (Pairwise Comparison CLIP) is visualized in Figure 1.

Motivated by our pairwise comparison-based finetuning, we develop a new inference mechanism, which we refer to as *comparative prompting*. This approach looks to improve downstream performance by incorporating prior knowledge in the form of text descriptions of the differences between classes. For instance, for a classification task between images of a crab and lobster, one can describe (or ask an LLM to describe) the following difference: "*Crabs have a rounded, flat body, while lobsters have a long body, large claws, and a pronounced tail.*" Our approach uses this comparative prompt to update and further separate the class prompts for these similar classes (see Figure 3).

We empirically demonstrate the many benefits of the improved reasoning ability from our finetuning approach on synthetic comparisons. First, we observe that PC-CLIP has the new capability of performing *difference-based classification*, or given a certain attribute (e.g., size and color) to correctly rank images of a pair by that attribute. In fact, PC-CLIP achieves significantly higher performance on this task (up to ∼14 points in absolute accuracy), while CLIP observes almost random performance. In addition, PC-CLIP has improved zeroshot classification performance, when using standard class prompts or more descriptive class descriptions, across a majority of downstream image classification tasks. These improvements hold even when compared to baselines of finetuning on the *exact same data from COCO* and an approach that leverages

*non-comparison-based* synthetic data from a LLM, demonstrating that benefits come from finetuning on comparisons. These improvements are even furthered when leveraging our comparative prompting technique with PC-CLIP, while using comparative prompting with CLIP features can exhibit large drops in performance. We also demonstrate that these benefits translate to cross-modal retrieval tasks, where we see improvements specifically on text-to-image retrieval. Finally, we demonstrate that PC-CLIP indeed satisfies a larger degree of geometric properties in embedding space, such as generating images of the result of arithmetic operations in text embedding space.

## 2 Related Works

**VLMs and Prompting** With the advent of VLMs such as CLIP (Radford et al., 2021) and ALIGN (Jia et al., 2021), a large body of work has studied ways to use these models. A main class of methods is prompting, which is a parameter-efficient technique to define classifiers given informative natural language descriptions of the classes of interest (Zhou et al., 2022). Some approaches leverage LLMs to extract additional information about classes instead of prompts that only use the class name, which achieves stronger performance and is perhaps more interpretable (Menon & Vondrick, 2022; Esfandiarpoor & Bach, 2024). Other approaches have learned these language descriptions both in continuous (Li & Liang, 2021) and discrete settings (Wen et al., 2023; Akinwande et al., 2023). Other recent work looks to extract particular concepts from pretrained models in a zeroshot fashion, with the goal of achieving more robust representations (Adila et al., 2023). Some works attempts to use VLMs to generate captions for images (Mokady et al., 2021) or the difference between images (Yao et al., 2022). Finally, an alternative class of VLMs is built from LLMs through visual instruction tuning (Liu et al., 2024). Our work is fundamentally different as it uses the reasoning abilities of LLMs to improve the geometry of *CLIP embeddings*.

**Finetuning** With the advent of these VLMs, many works have studied how to finetune these models for downstream tasks, instead of simply using fixed versions of these pretrained models. Many approaches study better ways to achieve more robust models via finetuning (Kumar et al., 2022; Wortsman et al., 2022). Prior work (Fan et al., 2023; Doveh et al., 2023) demonstrates that LLMs can be used to improve or diversify captions in pretraining data, leading to performance benefits. In addition, other work shows that given labeled downstream task data, a better way to perform finetuning is in line with the original pretraining objective (Goyal et al., 2023). Other work uses multiple LLM-generated class descriptions to improve few-shot finetuning (Feng et al., 2023). A relevant line of work is finetuning VLMs for image-difference captioning (Jhamtani & Berg-Kirkpatrick, 2018; Park et al., 2019; Guo et al., 2022; Hu et al., 2024), which looks to produce text descriptions of the difference between image pairs. While the underlying ideas in this line of work are similar, a key distinction is that we study the reverse problem, which is to study the benefits of incorporating such information into CLIP's embedding space.

**CLIP's Embedding Space** A large body of work has studied specific qualities of the learned embedding space of CLIP. A related work looks to better induce geometric properties in the resulting embedding spaces (Goel et al., 2022) through pairwise distances, although this does not directly address (LLM-generated) semantically meaningful differences. Other work finds that the embedding space of CLIP behaves like a bag of words (Yuksekgonul et al., 2022) and that the models lack the ability to bind particular attributes to instances (Lewis et al., 2022). Other work generates large synthetic datasets (via viewpoint modification, manual text generation via metadata) to improve VLMs abilities to reason about visual concepts and not individual objects (Cascante-Bonilla et al., 2023). Our work is related in that we demonstrate a failure of CLIP's embeddings, and we propose a new finetuning approach to address these issues.

**Using Language to Improve Performance and Interpretability** A wide variety of works have studied the use of natural language as human-interpretable explanations of model decisions. This has been studied and shown to improve both LLMs (Zhou et al., 2020; Lampinen et al., 2022a; Howard et al., 2023) and RL (Lampinen et al., 2022b). The most related setting is using these for VLMs, where prior work grounds explanations to modify the network's attention mechanisms (Petryk et al., 2022) or provide textual descriptions for specific, fine-grained regions of the image (Li et al., 2022b). Other work studies the setting of

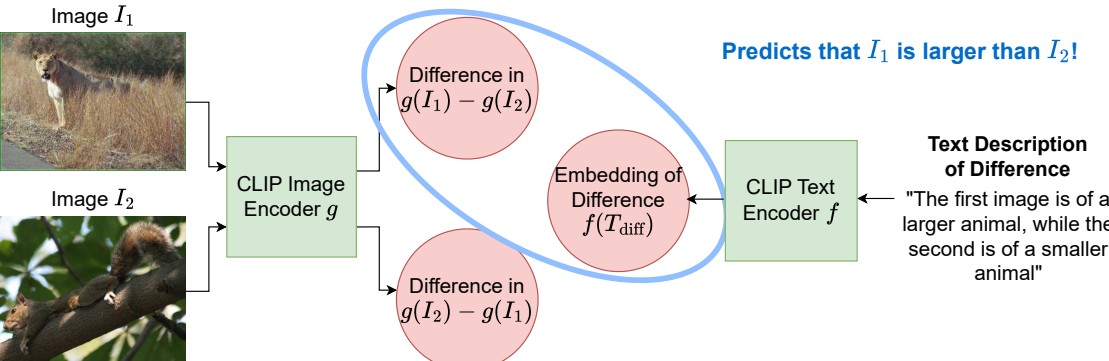

Figure 2: A visualization of difference-based classification. In this example, a VLM has a higher cosine similarity between $g(I_1) - g(I_2)$ and $f(T_{\text{diff}})$, which is represented by the blue oval. Thus, the model correctly predicts that image $I_1$ contains a larger animal (a lion) than image $I_2$ (a squirrel).

visual-textual entailment (Xie et al., 2019; Do et al., 2020), where the task is to determine whether the image entails the given textual description. On the contrary, we focus improving the embeddings of CLIP in terms of pairwise relationships between objects.

## 3 Methods

We now describe our approach to generate natural language descriptions of the difference between images with LLMs, and to finetune CLIP to better understand these meaningful differences. We then propose a technique to use the resulting model for general difference-based classification (i.e., ranking images in a pair correctly by a certain attribute), and comparative prompting, to leverage relational information between classes for improved downstream performance.

### 3.1 Generating Comparatives with LLMs

While CLIP models are trained via a contrastive learning objective, they perhaps surprisingly cannot perform analogies in their embedding space (Section 4.2). As such, we employ LLMs, which have been documented to exhibit an understanding of comparative information between different objects (Howard et al., 2023), to generate text supervision to explicitly encourage this behavior in VLMs and improve their ability to reason about differences in embedding space. We build off of image-caption datasets (Lin et al., 2014; Wah et al., 2011), which allow us to use LLMs to generate natural language descriptions of the difference between the images via the difference in their captions. This circumvents the requirement of acquiring costly human-labeled image differences (Yao et al., 2022).

Given a dataset of paired images and captions $\{(I_1, T_1), \ldots, (I_n, T_n)\}$, we use an LLM to generate a description of the difference in meaning between the two captions. This provides us with a source of weak supervision to incorporate explicit differences into the learned embeddings of the VLM. To generate these comparisons, we prompt an LLM with: "*What is the visual difference between an image with a description of {$T_1$} and an image with a description of {$T_2$}?*", along with a few prepended demonstrations of desired behavior. We automatically filter out low-quality generations (described in Appendix B.2) to produce a better, curated dataset of pairwise comparisons. We also provide ablations in Appendix C.4 where we report results without any filtering. Our strategy for eliciting this information from LLMs is outlined in entirety in Appendix B. In paired image-caption datasets, the captions can be rather succinct and may not capture the richness of the full image. Bridging the gap between the remaining information in the image and the caption, perhaps through using large multimodal models (Liu et al., 2024), is room for future work.

### 3.2 Incorporating Comparatives in VLMs

Now, we present our strategy to incorporate these LLM-generated pairwise comparisons into our VLMs through finetuning with a contrastive objective, as visualized in Figure 1. Given a pair of image-captions,

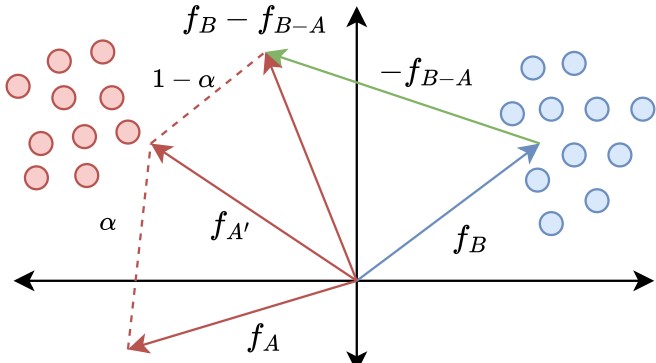

Figure 3: A visualization of **comparative prompting**. Arrows represent text embeddings of class (or difference) prompts, while circles represent image embeddings (red: class A, blue: class B). In this example, we can improve the inaccurate class prompt embedding $f_A$ by averaging it with $f_B - f_{B-A}$.

$(I_i, T_i), (I_j, T_j)$ and a corresponding text description of the difference between images $T_{i,j}$, we define our objective as follows

$$\min_{f,g} \ell\Big(g(I_1) - g(I_2), f(T_{1,2})\Big), \tag{1}$$

where $g, f$ represent our image and text encoders respectively. $\ell$ can represent any particular loss function. We primarily use the original CLIP contrastive loss (Equation (4)), but we also consider using the squared loss in Appendix C.2 and achieve similar results. In essence, this objective looks to align the difference between image embeddings to the corresponding embedding of the difference in captions produced by the LLM. As such, this better enforces geometric structure in CLIP's embedding space to reflect meaningful differences that are highlighted by an LLM.

### 3.3 Difference-Based Classification

PC-CLIP's improved ability to localize differences in its embedding space allows for the development of a more general classifier that reasons about differences, instead of solely relying on class name descriptions. We refer to this task as *difference-based classification*, or the ability to perform correctly determine an image in a pair of images that aligns with a certain attribute. For instance, if we are given an image of an elephant and a dog, we could reasonably ask and expect our model to know, "Which animal contained in the pair of images is larger?" Our difference-based classification task encompasses this question and other more general differences, such as color. This ranking capability can be used to build more general attribute-based classifiers, as in (Menon & Vondrick, 2022; Mazzetto et al., 2021b), or for retrieval or data curation, where the goal is to find a subset of images that better captures certain relevant properties for downstream tasks.

The loss of this task can be formally expressed on a pair of images $(I_i, I_j)$ and a corresponding text difference between the images $T_{i,j}$, such as the aforementioned question about size, as

$$\ell(f, g, I_i, I_j, T_{i,j}) = 1\Big\{\Big(g(I_i) - g(I_j)\Big) \cdot f(T_{i,j}) \geq \Big(g(I_j) - g(I_i)\Big) \cdot f(T_{i,j})\Big\}. \tag{2}$$

In essence, this task evaluates whether the model can properly order unlabeled images in relation to a particular attribute by using their difference in embedding space; this captures whether the difference in embedding space corresponds to meaningful concepts, such as size and color. A visualization of this task is given in Figure 2. We remark that some features such as size can be ambiguous, as it could refer to the inherent size of an object or the size of the object relative to the image; we focus on the former. We demonstrate in our experiments in Section 4.2 that CLIP performs poorly out-of-the-box on this task (achieving roughly random performance), reflecting that the contrastive objective does not suffice to successfully probe out relational information between images.

### 3.4 Comparative Prompting

PC-CLIP's improved embedding space also allows for a new type of inference to incorporate relational information between classes, which we refer to as comparative prompting. Given a prompt-based classifier, we can incorporate prior knowledge in the form of text descriptions of class-level differences to update our class prompts. As human-labeled image-level differences are expensive (and are potentially greater in cost to obtaining class labels), we focus on the setting where we have class-level differences, as we (or LLMs) can efficiently describe the differences between classes.

Let $A$ and $B$ represent two classes of interest, with embeddings $f_A$ and $f_B$ respectively. Given a language description of the difference between class $B$ and class $A$ (and an embedding of $f_{B-A}$), we can generate an updated class prompt $f'_A$ as follows:

$$f'_A := \alpha \cdot f_A + (1 - \alpha)(f_B - f_{B-A}), \tag{3}$$

where $\alpha$ is a hyperparameter that captures how much we rely on the comparison-based prompt. This captures our prior knowledge about differences in class descriptions by averaging the embedding $f_A$ with the difference in text embeddings $f_B - f_{B-A}$). Thus, if our original embedding of $A$ is inaccurate, this can be corrected if our embeddings of $(B - A)$ and $B$ are correct. A visual interpretation of this is provided in Figure 3. This exploits an asymmetry between the text representations of $f_{A-B}$ and $f_{B-A}$, which is perhaps lacking in CLIP as it has been shown to behave similarly to a bag-of-words (Yuksekgonul et al., 2022). Thus, our finetuning provides a simple solution to enable incorporating relational information into classification with contrastive-based VLMs.

## 4 Experiments

In evaluating our approach, we explore the following questions to understand the impacts of our finetuning. First, can PC-CLIP successfully perform difference-based classification, on different data distributions than our comparison-based finetuning dataset? Secondly, how does our finetuning impact our model's ability to perform zeroshot classification? Finally, does our finetuning generally improve the VLM's embedding space and its ability to perform arithmetic?

In our experiments, we first demonstrate that PC-CLIP can indeed perform difference-based classification on multiple downstream datasets with varying types of meaningful differences, while CLIP achieves almost random performance. Furthermore, we demonstrate that our comparative-based finetuning does not degrade standard zeroshot classification; rather, it improves performance with both simple class prompts and longer, descriptive prompts on a majority of image classification tasks. To control for having finetuned our model on COCO with synthetic LLM generations, we compare against (and outperform) additional baselines of directly *finetuning of CLIP on COCO* and finetuning on LLM-rewritten captions that have been *generated by the same LLM that we use to generate our comparisons*. This controls for the additional information from an external LLM and directly studies the benefits of incorporating comparative information. Finally, we demonstrate that PC-CLIP's text encoder is improved, better localizing class names with respect to their differences, and with better image generations of arithmetic operations in the text embedding space. This also manifests itself in larger classification performance gains with comparative prompting.

### 4.1 Experiment Details

**Generating Our Synthetic Dataset** To generate our PC-CLIP finetuning dataset of pairwise comparisons, we use LLaMA2-13B-chat-hf (Touvron et al., 2023). We find that this model gives more coherent descriptions of differences when compared to the base checkpoints that are not finetuned as chatbots. We generate comparatives on two datasets, COCO (Lin et al., 2014) and CUB-200-2011 (Reed et al., 2016). We report our primary results finetuning on comparisons derived from COCO, while we defer results with CUB to Appendix C.1. As the number of pairs scales quadratically in the dataset size, we create pairs (and their corresponding language differences) from 1000 randomly sampled images. We first perform a round of filtering based on simple heuristics on text language and structure, to filter out instances that have very low coherency (detailed in Appendix B.2), which results in a dataset of roughly 560,000 comparisons on COCO.

Table 1: Results on **difference-based classification** (e.g., binary classification among pairs of images determining which image is larger), which is described in detail in Section 3.3. Results are reported as mean ± standard error, when averaged over 5 seeds. We observe that CLIP and its finetuned version on COCO exhibit almost random performance, and PC-CLIP performs much better across all tasks.

| Method | AwA2 | CIFAR100 | CUB | Flowers102 |
|---|---|---|---|---|
| CLIP | $51.74 \pm 1.34$ | $54.92 \pm 1.11$ | $53.32 \pm 0.22$ | $52.97 \pm 2.12$ |
| CLIP (COCO FT) | $50.93 \pm 1.29$ | $55.12 \pm 1.27$ | $55.62 \pm 0.21$ | $53.62 \pm 2.02$ |
| CLIP (Rewrite FT) | $50.49 \pm 1.36$ | $55.52 \pm 1.37$ | $55.11 \pm 0.24$ | $54.18 \pm 2.01$ |
| **PC-CLIP** | $\mathbf{58.52 \pm 0.46}$ | $\mathbf{67.44 \pm 1.29}$ | $\mathbf{67.55 \pm 2.11}$ | $\mathbf{64.91 \pm 0.21}$ |

**Evaluation**   We evaluate our method on a variety of image classification tasks: CIFAR100 (Krizhevsky et al., 2009), Flowers102 (Nilsback & Zisserman, 2008), SUN397 (Xiao et al., 2010), EuroSAT (Helber et al., 2019), and CUB-200-2011 (Wah et al., 2011). We perform our difference-based classification on multiple datasets: Animals with Attributes 2 (AwA2) (Xian et al., 2018), CUB (Wah et al., 2011), CIFAR100 (Krizhevsky et al., 2009), and Flowers102 (Nilsback & Zisserman, 2008).

For difference-based classification tasks, instead of standard classification, we generate pairs of instances from different classes and an attribute that reflects a difference in these classes (e.g., "*The first image is larger*"). For AwA2, we have access to class-level binary attributes (e.g., fur, color, habitat) describing each class. We generate a string from the difference in these binary vectors between any two images from different classes. For CIFAR100, we use coarse-grained labels to infer information about relative *size* for classes. CIFAR100 contains the coarse-grained labels of "*large carnivores*", "*large omnivores and herbivores*", and "*small mammals*". Thus, we define a task of predicting which of the two images is larger, where one has a coarse-grained label containing "*large*" and while the other contains "*small*". On CUB, we have access to captions of each image, so we use LLaMA2 (Touvron et al., 2023) as before in generating differences. This most closely aligns with the same notion of differences as in pretraining, although there is a significant distribution shift as the images and captions are solely comprised of birds. Finally, on Flowers102, we can infer *color*, we generate a task for differentiating color between a group of yellow flowers ("*yellow iris*", "*daffodil*", "*sunflowers*", and "*goldenrod*") and a group of blue flowers ("*blue poppy*" and "*bluebells*"). Further details and examples of these differences for all datasets are given in Appendix A.4.

**VLM Finetuning**   In our experiments, we use a ViT-L/14 (Dosovitskiy et al., 2020) architecture with pretrained weights from Datacomp-1B (Gadre et al., 2023). In our finetuning, we update only the parameters of the text encoder. This allows us to precompute the image embeddings, which is significantly more computationally efficient. For our baseline of finetuning on COCO (and with LLM rewrites), we also update only the text encoder parameters, on the same 1000 examples from COCO used to generate our PC-CLIP dataset. We defer more details to Appendix A, and our code can be found here [1].

### 4.2   Difference-Based Classification Results

**PC-CLIP can perform difference-based classification, while CLIP cannot**   We report our results for our difference-based classification tasks in Table 1. We observe that CLIP struggles with this task, achieving almost random performance ($\sim 50\%$). Some intuition for this result is that CLIP has been primarily trained to align specific instances in its contrastive objective, and does not necessarily capture notions of semantic meaningful differences, leading to deficiencies on this and other related tasks. On the contrary, our finetuning helps improve performance by a large margin across all tasks. For instance, we see increases in performance by $\sim 14$ points in terms of absolute accuracy. This supports that PC-CLIP has a better alignment of the difference between image embeddings with more interpretable concepts such as size (e.g., CIFAR100) and color (e.g., Flowers102).

Table 2: (Top 4 rows): Comparison of PC-CLIP against CLIP features in terms of accuracy, when using standard class prompts (e.g., "*This is a photo of {class_name}*") for zeroshot image classification. (Bottom 4 rows): Comparison of these models in terms of accuracy when using comparative prompting, which leverages text descriptions of the *difference* between 3 pairs of highly confused classes. We bold the best-performing method when using standard or comparative prompting.

| Method | CIFAR100 | CUB | EuroSAT | Flowers102 | SUN397 |
|---|---|---|---|---|---|
| CLIP | 85.59 | **81.72** | 54.96 | 81.51 | 72.46 |
| CLIP (COCO FT) | 85.36 | 81.41 | 54.63 | 81.33 | 70.98 |
| CLIP (Rewrite FT) | 85.53 | 81.20 | 54.70 | 81.31 | 71.24 |
| **PC-CLIP** | **86.12** | 80.08 | **57.15** | **81.95** | **73.58** |
| CLIP + comp | 85.66 | **81.67** | 53.67 | 81.98 | 72.48 |
| CLIP (COCO FT) + comp | 85.34 | 81.27 | 56.78 | 81.92 | 70.98 |
| CLIP (Rewrite FT) + comp | 85.54 | 81.01 | 56.93 | 82.06 | 71.25 |
| **PC-CLIP + comp** | **86.08** | 80.01 | **60.30** | **82.78** | **73.64** |

Table 3: Comparing performance increase/decrease when using comparative prompting with CLIP and PC-CLIP on the classes that are updated with comparative prompts (i.e., 3 pairs of classes that are most commonly confused in standard prompting). We denote performance increases in red and decreases in blue. We bold the method that achieves the largest gain in performance.

| Dataset | CLIP | (+ comp) | PC-CLIP | (+ comp) |
|---|---|---|---|---|
| CIFAR100 | 68.33 | + 0.34 | 66.33 | **+ 1.34** |
| CUB | 59.43 | - 1.14 | 52.57 | **+ 0.57** |
| EuroSAT | 40.68 | - 3.40 | 44.75 | **+ 7.00** |
| Flowers102 | 44.07 | + 3.01 | 49.91 | **+ 9.38** |
| SUN397 | 74.73 | +0.18 | 72.33 | **+ 1.01** |

### 4.3 Classification Results

**PC-CLIP improves zeroshot classification performance on most downstream tasks** We evaluate the zeroshot classification performance of our methods using a class prompt (e.g., "*This is a photo of {class_name}*") for each of our target classes. As is done in standard practice, our classifier is defined by computing the cosine similarity between each text description of the target classes and making a prediction by taking the class with the largest cosine similarity. These experiments are primarily designed to assess whether our finetuning potentially degrades the original features learned during pretraining. We observe the *contrary*; our finetuning generally improves performance in terms of zeroshot prompting with class names (see Table 2). We observe that pretraining on LLM-generated comparatives on COCO improves performance on 4 out of the 5 downstream tasks that we consider. This supports that PC-CLIP not only allows new techniques such as difference-based classification, but also contains a useful signal for aligning its features with semantic classes.

**PC-CLIP observes larger and consistent performance gains with comparative prompting** As mentioned in Section 3.4, we can leverage information about the differences between pairs of classes to improve our standard class prompts. On these tasks, we generate this knowledge for both CLIP and PC-CLIP by looking at the confusion matrix of the zeroshot prompt-based classifier and selecting the 3 most confused class pairs. Then, given these pairs, we query GPT4 (OpenAI, 2023) for natural language descriptions that capture the difference between these different classes; for instance, "crab" vs. "lobster" classes, we prompt GPT-4 with: "In less than 30 words, what is the difference between a crab and lobster?" yielding: "Crabs have a rounded body, lobsters have large claws and tails." We present more details in Appendix A.2.

---

[1]https://github.com/dsam99/pc_clip

Table 4: Results when using LLM-extended class prompts for zeroshot image classification. We bold the best-performing method on each task. We observe that PC-CLIP achieves the highest performance on a majority of tasks.

| Method | CIFAR100 | CUB | EuroSAT | Flowers102 | SUN397 |
|---|---|---|---|---|---|
| CLIP | 84.32 | 81.41 | 59.04 | 81.23 | 69.98 |
| CLIP (COCO FT) | 84.30 | 81.79 | 57.81 | 81.30 | 69.57 |
| CLIP (Rewrite FT) | 84.4 | **82.21** | 59.11 | **81.48** | 68.77 |
| **PC-CLIP** | **85.56** | 79.65 | **59.59** | 79.54 | **73.05** |

We use these pairwise difference descriptions to update the class prompts, using the procedure in Equation (3). We remark that while this requires labeled data and thus is no longer truly zeroshot, this procedure does not require any training and is extremely easy to implement. In addition, our prior knowledge aligns with the pairs that are found in the confusion matrix; for instance, a confused pair of classes on the SUN dataset is "*kitchen*" and "*kitchenette*" and on the EuroSAT dataset is classes "*AnnualCrop*" and "*PermanentCrop*". Thus, we can instead generate pairs of confused classes through prior knowledge about semantically similar classes.

We observe that using our comparative prompting with PC-CLIP boosts or maintains performance on a majority of tasks, which is not the case when using comparative prompting with CLIP features (see the bottom two rows in Table 2). We remark that the gains in overall accuracy are not immediately apparent as we have only modified a small number of class prompts for tasks with large numbers of classes (e.g., SUN has 397 and CUB has 200). As such, we only observe slight gains as we have modified only a small fraction of the total classes. Thus, we also report the results for the accuracy on the subset of 6 classes (from 3 pairs) that are highly confused in Table 3. We remark that this subset of classes can be different for CLIP and PC-CLIP, although a majority are the same.

On this subset of highly confused tasks, the result is *much clearer*. We primarily focus on the columns labeled with (± comp), which denotes the change in performance when using comparative prompting. We observe that performing comparative prompting with CLIP helps performance on a few datasets, although it can also negatively impact performance (e.g., a large drop in accuracy on EuroSAT). However, comparative prompting more positively impacts performance for PC-CLIP, leading to a larger gain across all different tasks. Therefore, this supports that our finetuning enables the use of comparative prompting as a strategy to incorporate prior knowledge about class relations for downstream tasks.

To better understand the role of comparative prompting, we compare the tasks where it improves performance to those where performance roughly remains the same. Namely, we compute the improvement from comparative prompting and the confidence of the standard zeroshot classifier, which can be computed by looking at the difference in probability of the two most likely predicted classes (e.g., the margin). When computing the correlation between this confidence score and in the improvement from comparative prompting on the 5 datasets considered in the paper, we observe a correlation coefficient of -0.883. This strong negative correlation implies that comparative prompting helps in cases when the standard zeroshot classifier has low confidence; this is rather intuitive as this is the case when text-based class embeddings are likely the least accurate. This suggests possible refinements or improvements to comparative prompting as to only incorporate the comparative prompts on input data when the model is quite unconfident (e.g., in instances where the margin is below some threshold).

**PC-CLIP shows similar gains with longer class prompts.** Prior work demonstrates that prompting VLMs improves when using longer or more varied descriptions of classes (Menon & Vondrick, 2022). To evaluate how well our finetuning improves the performance of using longer and more descriptive prompts, we can swap standard class prompts for longer descriptions of the target classes. We again use LLaMA2 (Touvron et al., 2023) to generate these extended descriptions for class prompts; more details are described in Appendix A.3. Here, we remark that we see better performance if we perform weight-ensembling of PC-CLIP weights or COCO finetuned CLIP weights with the original CLIP weights, which is similar to prior work

Table 5: Average score (in a range from 1-5) of generations in text-to-image experiments from a sum of the embeddings of class names from CIFAR100 and attributes from AwA2, produced by a GPT-4o-mini judge, when averaged over 800 images. We observe that PC-CLIP (when used with SD-XL) produces images that better reflect the semantic arithmetic.

|  | CLIP | PC-CLIP |
|---|---|---|
| **LLM-Judge Score** | 1.75 | 2.33 |

Table 6: Results for cross-modal retrieval on Flickr30k. We bold the best-performing method on each task. We observe that PC-CLIP improves upon standard CLIP weights on image-to-text retrieval and outperforms all approaches on text-to-image retrieval.

|  | **Image-to-Text Retrieval** | | | **Text-to-Image Retrieval** | | | |
|---|---|---|---|---|---|---|---|
|  | **R@1** | **R@5** | **R@10** | **R@1** | **R@5** | **R@10** | **Mean Recall** |
| CLIP | **85.30** | 97.20 | 99.20 | 64.88 | 87.28 | 92.02 | 87.65 |
| **PC-CLIP** | 84.80 | **97.70** | 99.20 | **69.80** | **89.86** | **94.28** | **89.27** |

(Wortsman et al., 2022). Overall, PC-CLIP has stronger performance across a majority of datasets when using lengthier class descriptions (see Table 4).

## 4.4 Cross-Modal Retrieval Results

We also evaluate the zeroshot performance of our finetuned model in performing cross-modal retrieval on the test set of the Flickr30k dataset (Young et al., 2014). We note that we truncate captions to the maximum sequence length of the CLIP text encoder, which is 77 tokens. We report both image-to-text and text-to-image retrieval results in Table 6, finding that PC-CLIP maintains roughly the same image-to-text retrieval performance while greatly improving upon text-to-image retrieval when compared to the standard CLIP weights.

## 4.5 Evaluating the Quality of Learnt Embeddings

**Embedding Arithmetic with Text-to-Image Generation**   To evaluate the ability of a text encoder to perform arithmetic in text embedding space is to visualizing the resulting summation of embeddings through an existing text-to-image generation model, such as Stable Diffusion (Rombach et al., 2022; Podell et al., 2023). Here, we can directly swap the text encoder in Stable Diffusion XL (Podell et al., 2023) with one that has been finetuned with our PC-CLIP objective. We remark that this reflects a different CLIP model architecture (namely a ViT-G/14), as this is the text embedding used in Stable Diffusion XL.

We consider a task of producing images from a combination of an object corresponding to a class from CIFAR100 and a specific color attribute (taken from the AwA2 dataset). Therefore, we take the summation of the embeddings that correspond to the CIFAR100 class and the color attribute and feed this into a text-to-image generation model to visualize the result of the semantic arithmetic. More details about this generation process are outlined in detail in Appendix D.2. Note that we do not present this as a specific approach to perform compositional image generation (Liu et al., 2022) (as we could easily lengthen the original prompt), but rather, we use this to evaluate PC-CLIP's improved ability to perform text embedding arithmetic.

To quantitatively evaluate how well the generated images match the sum of two text prompts, we can evaluate them using a large multimodal model as a judge. Specifically, we use GPT-4o-mini to judge the quality of generations on a scale from 1 to 5, with 5 being of highest quality. We provide the specific scoring prompt used in Appendix B.2. We remark that finding metric to evaluate text-image alignment is an open research question. We evaluate a set of images generated to represent CIFAR100 classes while adding in the text

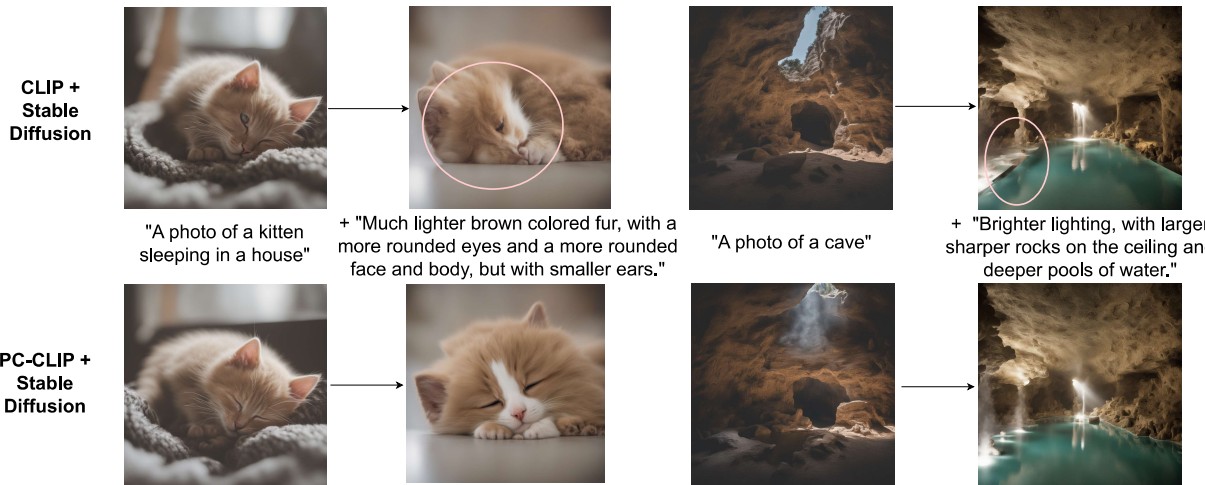

Figure 4: Visualization of the text encoder of PC-CLIP as we add a descriptive statement through a text-to-image generation model (Stable Diffusion XL (Podell et al., 2023)). Areas that are circled in light red denote visual inconsistencies in the generated images when using CLIP.

Table 7: Comparing the text encoders of PC-CLIP and CLIP. (First two columns: Comparison) We report the cosine distance between the difference in class prompts embeddings and LLM-generated comparison embedding (i.e., $d(f_A - f_B, f_{A-B})$), averaged over the 3 most confused pairs of classes in classification. (Last two columns: Reverse Comparison) We report the cosine distance from the negative difference (e.g., $f_{B-A}$ instead of $f_{A-B}$) of class embeddings to the comparison embedding. ($\uparrow$) denotes larger is better, and ($\downarrow$) denotes that smaller is better.

|  | Comparison ($\downarrow$) | | Reverse Comparison ($\uparrow$) | |
| --- | --- | --- | --- | --- |
|  | **CLIP** | **PC-CLIP** | **CLIP** | **PC-CLIP** |
| CIFAR100 | 1.04 | **0.92** | 0.96 | **1.08** |
| CUB | 1.19 | **1.07** | 0.81 | **0.93** |
| EuroSAT | 0.92 | **0.73** | 1.08 | **1.27** |
| Flowers102 | 1.08 | **0.99** | 0.92 | **1.01** |
| SUN397 | 1.06 | **0.90** | 0.94 | **1.10** |

embedding of attributes from AwA2. We observe that PC-CLIP, when used with Stable Diffusion, produces images that achieve a higher CLIP score than the original CLIP embedding, reflecting better arithmetic properties in embedding space.

We qualitatively observe that adding in the text embedding of (especially long) comparison-based descriptions to original text prompts leads to slightly more visually consistent generations with our text (see Figure 4). In the provided examples, these comparison-based descriptions negatively impact the visual coherence of generations from CLIP + Stable diffusion (regions circled in red in Figure 4), while PC-CLIP + Stable Diffusion much better captures the text from the comparison-based additions in the generated images. We provide more examples and a more in-depth discussion in Appendix D.3.

**PC-CLIP better localizes classes and their differences** We consider the same notion of language descriptions of pairwise class differences as in our comparative prompting. Here, we assess text encoder quality as $d(f_A - f_B, f_{A-B})$, which measures the distance between the difference in our model's embedding and our model's embedding of the semantic (LLM-generated) difference. We argue that this is a reasonable metric, as a desirable property of our model is to capture nice geometric properties, such as obeying arithmetic operations in the embedding space.

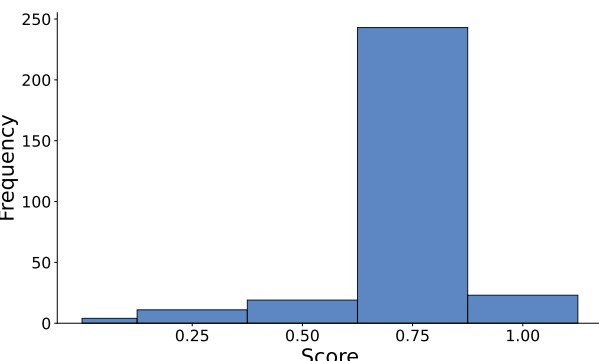

Figure 5: A visualization of scores produced by GPT-4o-mini in judging the quality of our synthetic data generations. 300 examples are judged on a scale in [0, 1], where 1 reflects higher quality.

We report the distance for both our model and the standard CLIP features in Table 7). In these experiments, we report the average distances in embedding space over the pairs of confused classes in each downstream task from our comparative prompting. As mentioned previously, for the text description of the difference between these confused classes (i.e., the inputs to extracting our difference embedding $f_{A-B}$), we use GPT-4 to generate these comparisons, using the exact prompt provided in Appendix A.2. We report the cosine distance as our distance function $d$.

We observe that the text encoder is significantly improved across all datasets. The first two columns illustrate that it better aligns the difference in text embeddings with the corresponding actual language description of the difference. The last two columns demonstrate that our finetuning does not simply collapse the representation space; the negative difference in class prompt embeddings is further away from the description of the difference. This better localization provides a likely explanation for better performance in difference-based classification and from comparative prompting.

### 4.6 Analysis of Synthetic Data Quality

We also provide an analysis of the quality of our synthetically generated differences in Figure 5. We pass the corresponding images, captions and our synthetically generated description of the difference in images to GPT-4o-mini, which has been trained to handle multimodal inputs and perform general reasoning behavior. To measure the quality of our generation, we use GPT-4o-mini as a judge to provide a score in [0, 1] of our generation, where 1 corresponds to a description that contains all relevant differences and where 0 corresponds to no meaningful information. We detail the specific scoring prompt used in Appendix B.2.

We generally find that our generations are of fairly high quality, with most having a score of 0.75. We also analyze the generated explanations from the LLM judge to determine what often causes these imperfect scores. These generations are often imperfect as they are built from caption information only, and the majority of generations with a score of 0.75 miss subtle differences that are contained within the background (e.g., information that is not generally contained in the COCO captions). For the small fraction of examples with scores of 0.50 or lower, we observe that there are key details missing in the original COCO captions.

## 5 Discussion

We propose a method to improve CLIP's embedding space by generating language descriptions of the difference between images and using this dataset to improve the joint embedding space of CLIP to reflect more interpretable differences between classes, such as size and color. We demonstrate that our finetuning enables the ability to perform general difference-based classification while generally improving or maintaining standard zeroshot prompting performance with our updated VLM. With other simple metrics and text-to-

image visualizations, we find that the embedding space of PC-CLIP indeed better captures meaningful notions of differences, which can later improve many downstream applications that build on top of CLIP embeddings.

A fundamental limitation of our method is that we rely on the ability of LLMs to generate these image comparisons from imperfect information (i.e., only the text caption). These models can sometimes leverage general information that does not apply to particular images, as well as have poor responses due to issues such as hallucinations (Zhang et al., 2023). This can likely be improved by the advent and usage of large multimodal models that exhibit both image and language understanding (OpenAI, 2023; Liu et al., 2024). In addition, these models themselves are often prone to hallucination, which can lead to poor-quality synthetic data.

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

## A Additional Experiment Details

We now present additional details in our experimental setup.

### A.1 Hyperparameters

**PC-CLIP COCO Finetuning**   We finetune CLIP with our comparative-based objective on COCO using the following hyperparameter values:

- $\tau = 1.0$ as our temperature value in the contrastive loss function

- learning rate of $10^{-8}$, with an exponential scheduler with $\gamma = 0.9$

- 20 epochs of finetuning

- batch size of 512

For our experiments using the MSE as our loss function, we instead only train for 5 epochs, as this different objective can significantly degrade the quality of the learned features. For weight ensembling, we simply average the two sets of weights.

**CLIP COCO Finetuning**   We finetune CLIP on COCO with its original contrastive objective with ground truth captions and LLM-rewritten synthetic captions using the following hyperparameter values:

- $\tau = 1.0$ as our temperature value in the contrastive loss function

- learning rate of $10^{-6}$, with an exponential scheduler with $\gamma = 0.9$

- 10 epochs of finetuning

- batch size of 128

We choose a smaller learning rate given that there are significantly fewer individual data than the number of pairs in our comparative task (although they are using the same number of COCO annotated examples).

**Comparative Prompting**   In our comparative prompting, we have one parameter $\alpha$, which controls how much we adjust our class prompts with the class-level comparative prompt. For all tasks, we evaluate with $\alpha \in \{0.5, 0.7, 0.9\}$. In our results, we report $\alpha = 0.9$, which seems to be the best performing across all tasks for both our finetuned model and the vanilla CLIP weights.

**PC-CLIP CUB Finetuning**   We finetune CLIP on the CUB dataset with our comparative-based objective using the following hyperparameter values:

- $\tau = 1.0$ as our temperature value in the contrastive loss function

- learning rate of $10^{-8}$, with an exponential scheduler with $\gamma = 0.9$

- 20 epochs of fine-tuning

We remark that on CUB, we generate comparatives on pairs generated from 750 instances, leading to a pretraining dataset of half the size of that of COCO.

### A.2 Generating Comparative Prompts

In generating our comparative prompts, we compute the confusion matrix to get the 3 most commonly confused pairs of classes. Then, given these classes, we generate a comparative that describes the difference between the pair of classes by prompting GPT4 (OpenAI, 2023) with:

*In less than 30 words, what is the description of the visual difference (e.g., in terms of color or shape) between an image of {class_1} and an image of {class_2}?*

We include the generated responses in our code base. This procedure of comparative prompting is similar to leveraging prior information (as is commonly done in few-shot or semi-supervised learning (Wang et al., 2020; Pukdee et al., 2023)), as many of these confused classes are similar in semantic meaning. On the considered datasets, the commonly confused classes are:

- CIFAR100: "*crab*" and "*lobster*", "*maple tree*" and "*oaks*", "*porcupine*" and "*shrew*"

- CUB: "*Le Conte's Sparrow*" and "*Nelson's Sharp-tailed Sparrow*", "*Chuck-will's widow*" and "*Nighthawk*"; "*Geococcyx*" and "*Sayornis*"

- EuroSAT: "*PermanentCrop*" and "*AnnualCrop*", "*SeaLake*" and "*PermanentCrop*", "*Pasture*" and "*PermanentCrop*"

- Flowers102: "*Petunia*" and "*Mexican Petunia*", "*Bishop of Llandaff dahlia*" and "*orange dahlia*", "*thorn apple*" and "*balloon flower*"

- SUN: "*kitchen*" and "*kitchenette*", "*scene restaurant*" and "*bistro*"; "*bedroom*" and "*hotel room*"

Many of these pairs, such as "*Petunia*" and "*Mexican Petunia*", "*kitchen*" and "*kitchenette*", and "*crab*" and "*lobster*", capture semantically similar classes, where we expect that more fine-grained descriptions can help us better perform classification. For these particular pairs, the comparative prompts are given by

- "*Petunia*" and "*Mexican Petunia*": "*Petunia flowers have funnel-shaped blooms, often with a broad range of colors; Mexican Petunia bears trumpet-shaped flowers, typically in violet or blue hues.*"

- "*kitchen*" and "*kitchenette*": "*A kitchen is typically larger with full-sized appliances; a kitchenette is smaller, with compact appliances and limited space.*"

- "*crab*" and "*lobster*": "*Crabs have a rounded, flat body with two claws, while lobsters have a long body, large claws, and a pronounced tail.*"

These comparatives capture more specific differences between these class labels, and, thus, can be helpful for prediction tasks by separating the original class prompts for the these classes.

### A.3 Generating Extended Class Descriptions

For our extended class description experiments, we also use LLaMA2-13B to generate a longer description of each class. We prompt the LLM with the following text:

*Q: What is a longer description of the visual features of the class "dog"?*
*A: Dogs possess four legs with distinctive paws, sharp teeth, keen senses, expressive eyes, and a snout, all contributing to their unique and diverse physical appearances.*
*Q: What is a longer description of the class "class_name"?*

*A:*

Again, we observe that by providing a demonstration (which was generated via GPT4), the quality of the output is more coherent and consistent across different classes. We then use the outputted response (up to 80 tokens) as a replacement for standard class prompts. These class prompts capture a wider variety of discriminative factors, which can aid in classification performance, which is noted by prior work (Menon & Vondrick, 2022). This indeed can be used in combination with our comparative prompting scheme, and generating more discriminative original class prompts is orthogonal to our difference-based approaches.

## A.4 Difference-based Classification Details

On AwA2, CIFAR100, and Flowers102, we evaluate our approaches on pairs generated from 100 randomly sampled images that are from different color or size groups. On CUB, we evaluate performance on a total of 5000 pairs. In our difference-based classification task, we generate our pairwise differences as follows on the following datasets:

**AwA2** On AwA2, we have access to the class-level binary attribute vectors for each image. For each unlabeled pair of images, we compute the difference in these binary attribute vectors, similar to previous work in using these attributes for classification (Mazzetto et al., 2021b;a; Sam & Kolter, 2023). In other words, we construct two sets of attributes: (i) those that are contained in the first image and not the second ($A_1$) and (ii) those that are contained in the second and not the first ($A_2$). Then, we can construct our text difference as

$T_{\text{diff}} \coloneqq$ *The first image has attributes of {$A_1$}, while the second image has attributes of {$A_2$}.*

where $A_1$ and $A_2$ are the string names of the attributes (e.g., "*brown*", "*furry*", "*active*", etc.) joined as a comma-separated list. We also remark that the AwA2 dataset has a large number of unhelpful attributes, which are not necessarily useful in terms of visual descriptions. Therefore, we filter out a set of unhelpful attributes (e.g., "*insects*" or "*fish*" when describing the animal's diet, "*smelly*", "*stalker*", etc.).

**CIFAR100** As previously mentioned, we group a few sets of classes into "*large animals*" and "*small animals*", through the coarse-grained labels from the dataset. Then, we generate pairs of data where one image comes from a group of large animals and the other comes from small animals. For a pair of images ($I_1, I_2$), if the first image comes from the group of large animals, our difference is given by:

$T_{\text{diff}} \coloneqq$ *The first image contains a larger animal, while the second contains a smaller animal,*

and if the first is from the group of smaller animals, then our difference is given by:

$T_{\text{diff}} \coloneqq$ *The first image contains a smaller animal, while the second contains a larger animal.*

**CUB** On the CUB dataset, we generate our differences in the same fashion as we have for the comparatives in our pretraining data (see Appendix B). Here, we precompute differences across 400 randomly sampled instances in the test dataset, and we randomly sample 5000 pairs (and differences) for our classification task. Thus, we would perhaps intuitively expect increased performance, although we do remark there is still a significant notion of a distribution shift when pretraining on COCO and then transferring the learned features to the task over CUB.

**Flowers102** On the Flowers102 dataset, we generate our differences in terms of color by grouping a small set of classes into "*yellow flowers*" and "*blue flowers*", as previously mentioned. For any pair of images $(I_1, I_2)$, if the first image comes from the group of yellow flowers, our difference is given by:

$T_{\text{diff}} \coloneqq$ *The first flower is yellow, while the second is blue,*

and if the first is from the group of smaller animals, then our difference is given by:

$T_{\text{diff}} \coloneqq$ *The first flower is blue, while the second is yellow.*

## A.5 Compute Resources

We compute our LLM-generated comparatives using a single A100 GPU or 2 A6000 GPUs, and the total process requires approximately 30 GPU hours. In our finetuning of the text encoder of PC-CLIP, we use a single A100 or A6000 GPU, which takes roughly 12 GPU hours to train for 20 epochs over our set of roughly 560,000 comparatives and pairs of images on COCO.

## B  LLM Generation Details

We now present our procedure to automatically generate natural language pairwise comparisons between images using LLaMA2-13B (Touvron et al., 2023) on image-caption paired pretraining data. We specifically use the LLaMA2-13B-chat-hf checkpoint, as we have found that this produces significantly more coherent results than the LLaMA2-13b checkpoint without any finetuning on human feedback. As mentioned in the main body of the paper, we primarily consider continuing pretraining with pairwise comparatives on COCO (Lin et al., 2014). We also discuss pretraining on comparatives on another dataset CUB-200-2011 (Wah et al., 2011) in the Appendix, which is more domain-specific but contains more semantically similar classes that can result in more meaningful pairwise differences. The specific prompting strategy to generate comparisons for each of these datasets is given below.

### B.1  Prompting Strategies

**COCO Dataset** To generate a comparative for a pair of image-text pairs $(I_i, T_i)$, $(I_j, T_j)$ on the COCO dataset, we prompt our LLM with the following prompt:

---

**Prompt Details**

Q: What is the visual difference between an image captioned with "a photo of a black, small cat" and an image captioned with "a photo of a large, white dog"?
A: The cat is smaller and is the color black, while the dog is larger and is white.

Q: What is the visual difference between an image captioned with "a photo of a large, white dog" and an image captioned with "a photo of a black, small cat"?
A: The dog is larger and is the color white, while the cat is smaller and black.

Q: What is the visual difference between an image captioned with "a photo of a house" and an image captioned with "a photo of an airport"?
A: The house contains furniture and homely decorations, while the airport is much larger and a public space.

Q: What is the visual difference between an image captioned with "a photo of an airport" and an image captioned with "a photo of a house"?

---

---

A: The airport contains travelers and airplanes and is a public space, while the house is smaller and is a private space.

Q: What is the visual difference between an image captioned with "$\{T_1\}$" and an image captioned with "$\{T_2\}$"?
A:

---

**CUB-200-2011 Dataset**   On the CUB dataset, we use the following prompt:

---

### Prompt Details

Q: What is the visual difference between an image with a description of "a grey bird with small wings and a yellow beak" and an image with a description of "a blue bird with large wings and a brown beak"?
A: Difference in color and size of the wings. One is grey and has small wings and a yellow beak, while the other is blue and has large wings and a brown beak.

Q: What is the visual difference between an image with a description of "a brown bird with an orange beak" and an image with a description of " a black bird with yellow beak"?
A: The color of the body and the beaks. One has a brown body and orange beak, while the other is black with a yellow beak.

Q: What is the visual difference between an image with a description of "$\{T_i\}$" and an image with a description of "$\{T_j\}$"?
A:

---

We observe that the demonstrations of questions and answers significantly improve the quality and consistency of the format of responses, which is in line with results from in-context learning (Min et al., 2022). For both tasks, we use the first 80 tokens produced by the language model as our comparative. We then pass these responses through a lightweight filtering process to remove or clean low-quality generations.

## B.2   Filtering Procedure

While our prompting strategy overall leads to higher-quality generations, there are still many low-quality responses. We employ the following filtering strategy:

- We filter out responses containing "*#include*" and "*#define*"; this captures the failure mode of LLaMA2 that generates responses of code and has no underlying semantic meaning or relation to the images in question.

- We filter out responses containing 8 repeated newline characters; this captures the failure mode of LLaMA2 that only generates newline characters.

In addition, we also use heuristics to remove parts of the generated responses to improve quality. For instance, we ignore any characters after instances of "Q:", which indicates that LLaMA2 is generating another question and answer, that is not necessarily related to the pair of instances $(I_i, T_i), (I_j, T_j)$. Similarly, we ignore all characters including and after "Note:", which is some generic disclaimer outputted by the model, which is again not related to our input instances. Overall, this filtering procedure reduces from a total of 1,000,000 generations to a filtered set of 560,000 generations for the COCO dataset. We remark that we generated these heuristics from a quick pass through a small subset of the LLM responses, although it can likely be improved with a more thorough study of a larger number of responses.

### B.3   Synthetic Data Scoring Details

We now present details used in our results for judging the quality of our synthetic generations of differences using GPT-4o-mini. We use the following scoring prompt, where we plugin values for caption1, caption2, and comparative below:

---

**Prompt Details**

I'm showing you two images with their captions and a comparative description.

Image 1 caption: {caption1}
Image 2 caption: {caption2}
Image 1: {image1}
Image 2: {image2}
Comparative description: {comparative}

Please provide:
1. A brief description of what you see in Image 1 (2–3 sentences)
2. A brief description of what you see in Image 2 (2–3 sentences)
3. Evaluate the quality of the comparative description on a scale of 0 to 1, where:

- 0 means the description fails to accurately capture meaningful differences

- 0.25 means the description captures some differences but misses some details or hallucinates new objects

- 0.5 means the description captures some differences but is partly inaccurate

- 0.75 means the description captures most differences with minor inaccuracies

- 1 means the description perfectly captures the key differences

4. Briefly explain your reasoning for the score (2-3 sentences). Format your response as valid JSON with keys: `image1_description`, `image2_description`, `score`, `reasoning`

---

As the API also receives images, we pass in the corresponding pair of images from COCO.

### B.4   Text-to-Image Generation Scoring Details

We now present the exact prompt used for our GPT-4o-mini judge for assessing the quality of text-to-image generations. We use the following scoring prompt, plugging in values for `prompt` and `image`.

---

**Prompt Details**

You are an expert image judge. Please judge how well the corresponding generated image follows the specified prompt, which is an attribute and a particular object.
First, think step-by-step about how well the image matches the prompt.
Then respond **only** with a JSON object containing:

- A string field `"explanation"` with a brief description of your reasoning (less than 20 words)

- A numeric field `"score"` between 1 and 5

**Interpretation of scores:**

- 1: Image is unrelated to the prompt

- 3: Image contains the object but ignores the attribute/descriptor (partial match)

---

- 5: Image incorporates both the attribute and object correctly

**Example:**
```
{
  "explanation": "I see a red apple on a tree but no spots, so partial match.",
  "score": 3
}
```
**Prompt:**
Generate an image of the following: {prompt}
Image: {image}

## C    Additional Experiments

We now present additional experiments with finetuning on different pretraining datasets of comparatives and with different losses in our fine-tuning objective for PC-CLIP.

### C.1    Other Pretraining Datasets

We also experiment with finetuning on comparatives generated from the CUB-200-2011 dataset (Wah et al., 2011). Here, we hypothesize that the differences between images are potentially more meaningful than on COCO, as it is much easier to reason about the differences between types of birds; the differences are more constrained to particular attributes such as size, color, and other attributes inherent to birds. Thus, more comparisons can be in relative terms (as it is hard to relate significantly different classes such as giraffes and houses from COCO).

Table 8: Experiment on alternating the underlying dataset for our comparative-based finetuning process for PC-CLIP. We report *difference-based classification* accuracy across multiple tasks, averaged over 5 random seeds.

| Dataset | PC-CLIP (COCO) | PC-CLIP (CUB) |
|---|---|---|
| AwA2 | $58.52 \pm 0.46$ | $46.84 \pm 0.89$ |
| CIFAR100 | $67.44 \pm 1.29$ | $83.30 \pm 1.07$ |
| CUB | $67.55 \pm 2.11$ | $69.09 \pm 2.50$ |
| Flowers102 | $64.91 \pm 0.21$ | $72.41 \pm 0.13$ |

Table 9: Experiment on alternating the underlying dataset for our comparative-based finetuning process. We report *standard zeroshot prompt* accuracy across multiple downstream image classification tasks. We observe that finetuning on CUB is slightly worse than on COCO.

| Dataset | PC-CLIP (COCO) | PC-CLIP (CUB) |
|---|---|---|
| CIFAR100 | 86.12 | 85.70 |
| CUB | 80.08 | 78.12 |
| EuroSAT | 57.15 | 55.07 |
| Flowers102 | 81.95 | 78.91 |
| SUN | 73.58 | 70.68 |

We observe that continuing pretraining with comparatives on the CUB-200-2011 dataset can lead to better performance in terms of difference-based classification results (see 8). For instance, we see better performance on discerning size on CIFAR100 and LLM-generated descriptions on CUB. This is somewhat intuitive, as the differences incorporated in the model are more in line with the tasks on these two datasets. However,

we note that there is worse performance than when pretraining on COCO in terms of standard prompting (and even sometimes when compared to the original VLM's weights), which is shown in Table 9. Somewhat surprisingly, we do not see a large performance boost when performing downstream zeroshot classification on CUB. We remark that the pretraining objective does not take into account the original caption information (except in a very indirect fashion through the LLM-generated comparative), and this provides a potential explanation for the lack of performance gain.

Overall, these experiments highlight that the comparative dataset does play an important role in the impact on downstream model performance. The nature of the pretraining dataset determines the generated differences from the LLM, as in the case of CUB-200-2011, differences are primarily in terms of size and color. This translates to a better understanding of these particular differences, while on COCO, we observe much more varied objects, which likely contributes to the better performance on a larger variety of classification tasks when pretraining on COCO. An interesting area for future work could address constructing a mixture of datasets of differences, which could be generated over a union of different pretraining datasets to capture more fine-grained notions of differences and maintaining diversity in image pairs. This is related to work in selecting relevant tasks via our domain knowledge, which can be thought of as defining a useful prior (Sam et al., 2024).

## C.2   Other Finetuning Objectives

The loss that we consider in our objective for PC-CLIP is given by the standard contrastive learning loss used in training CLIP (Radford et al., 2021):

$$\ell(X, Y) = -\frac{1}{2} \sum_{(x,y)} \left( \log \frac{\exp(x^\mathsf{T} y / \tau)}{\sum_i \exp(x_i^\mathsf{T} y / \tau)} + \log \frac{\exp(x^\mathsf{T} y / \tau)}{\sum_j \exp(x^\mathsf{T} y_j / \tau)} \right), \tag{4}$$

where $X, Y$ are a batch of normalized image and text (difference) embeddings, and where $\tau$ is the temperature hyperparameter. As previously mentioned, we could also consider using the mean-square error as a metric instead of CLIP's contrastive loss. This is given by

$$\ell_{mse}(X, Y) = \sum_i \left( x_i - y_i \right)^2, \tag{5}$$

where again $X, Y$ represent batched differences in image embeddings and batched text embeddings of LLM-generated differences.

Table 10: Using MSE as our finetuning objective (MSE), instead of the standard contrastive loss for PC-CLIP. We report *difference-based classification* accuracy across a variety of tasks.

| Dataset | PC-CLIP | PC-CLIP (MSE) |
|---|---|---|
| AwA2 | $58.52 \pm 0.46$ | $57.08 \pm 0.42$ |
| CIFAR100 | $67.44 \pm 1.29$ | $68.03 \pm 1.24$ |
| CUB | $67.55 \pm 2.11$ | $67.27 \pm 2.05$ |
| Flowers102 | $64.91 \pm 0.21$ | $65.36 \pm 0.19$ |

We empirically observe that using a squared loss in our objective achieves roughly similar performance on both difference-based classification and on standard prompting (Table 10 and 11). In general, it seems that with the contrastive loss, zeroshot classification performance is marginally better, while difference-based classification is marginally worse.

## C.3   Image-Difference Captioning Results

We present results for experiments on image-difference captioning and retrieval, following the experimental guidelines in the work of Guo et al. (2022). Specifically, we evaluate the performance of PC-CLIP in comparison to CLIP when integrated into their CLIP4IDC pipeline. PC-CLIP consistently demonstrates

Table 11: Using MSE as our finetuning objective (MSE), instead of the standard contrastive loss for PC-CLIP. We report *standard zeroshot propmting* accuracy across a variety of image classification tasks.

| Dataset | PC-CLIP | PC-CLIP (MSE) |
|---|---|---|
| CIFAR100 | 86.12 | 86.08 |
| CUB | 80.08 | 80.36 |
| EuroSAT | 57.15 | 55.59 |
| Flowers102 | 81.95 | 81.79 |
| SUN | 73.58 | 73.68 |

improved performance for both retrieval (i.e., identifying the pair of images described by a textual difference) and captioning (i.e., describing the difference between two images). The results are summarized in Tables 12 and 13. This also further improves upon baselines taken from the work of Guo et al. (2022).

Table 12: Spot-the-Difference text to image-pair retrieval results (R@5 and R@10) with IDC using CLIP pretrained weights and PC-CLIP.

| | R@5 | R@10 |
|---|---|---|
| IDC + CLIP | 3.0 | 3.7 |
| **IDC + PC-CLIP** | **3.6** | **5.2** |

Table 13: Spot-the-difference captioning results with IDC using CLIP pretrained weights and PC-CLIP. The reported metrics are B (BLEU-4), M (METEOR), C (CIDEr), R (ROUGE).

| Model | B | M | C | R |
|---|---|---|---|---|
| IFDC (Huang et al., 2021) | 8.7 | 11.7 | 37.00 | 29.90 |
| VACC (Shi et al., 2020) | 8.1 | 12.5 | 34.5 | 32.10 |
| IDC + CLIP | 10.61 | **12.82** | 41.17 | 32.96 |
| **IDC + PC-CLIP** | **10.96** | **12.82** | **43.09** | **33.24** |

### C.4  Ablation on Filtering LLM Generations

We conducted an ablation study to evaluate the impact of using the full, unfiltered dataset for fine-tuning PC-CLIP, and the robustness of our finetuning to noise in the LLM generated differences. The unfiltered dataset includes the original $\sim 1$M examples, compared to the filtered set of $\sim 560$k examples from which $\sim 440$k examples were removed. We observe that while PC-CLIP with filtering achieves the strongest performance on a majority of tasks, PC-CLIP Unfiltered outperforms vanilla CLIP weights on 4 of the 5 tasks, and even outperforms PC-CLIP with filtering on one task. This supports that our finetuning method is robust to noise in the LLM generations.

### C.5  Results for Natural Distribution Shifts

In addition to the zeroshot results on a wide variety of various downstream tasks, we also add an additional evaluation on natural distribution shifts (e.g., ImageNet-A (Hendrycks et al., 2021b) and ImageNet-R (Hendrycks et al., 2021a)). We observe that PC-CLIP has slight performance improvements over the CLIP baseline (Table 15) for natural distribution shifts as well as stronger distribution shifts considered in many of the zeroshot tasks (e.g., EuroSAT).

Table 14: Ablation on PC-CLIP trained with filtered and unfiltered LLM generated data. We bold the best-performing method and underline the second best-performing method.

| Model | CIFAR-100 | CUB | EuroSAT | Flowers | SUN |
|---|---|---|---|---|---|
| CLIP | 85.59 | **81.72** | 54.96 | 81.51 | 72.46 |
| PC-CLIP Unfiltered | 85.81 | 80.46 | **58.81** | 81.57 | 73.11 |
| PC-CLIP | **86.12** | 80.08 | 57.15 | **81.95** | **73.58** |

Table 15: Performance on natural distribution shift benchmarks (ImageNet-R and ImageNet-A).

| Model | ImageNet-R | ImageNet-A |
|---|---|---|
| CLIP | 69.07 | 90.33 |
| PC-CLIP | **69.2** | **90.47** |

## C.6 Linear Probe Results

To evaluate the performance of the learned features in PC-CLIP, we run experiments doing linear probing with labeled data from the downstream task. As we have primarily run experiments with PC-CLIP where we only update the text encoder, we also perform full finetuning to update the image encoder, so that we can evaluate the linear probe performance. For each task, we consider using 100 labeled instances per class. These results (Table 16) indicate that while PC-CLIP shows slight improvements in vision embeddings, the most significant benefits arise from improvements in the text encoder. This aligns with prior work highlighting limitations in CLIP's text embedding space (Yuksekgonul et al., 2022).

Table 16: Linear probing results on vision embeddings produced by CLIP and PC-CLIP.

| Model | CIFAR-100 | CUB | EuroSAT | Flowers | SUN | ImageNet-A | ImageNet-R |
|---|---|---|---|---|---|---|---|
| CLIP | **90.78** | **88.51** | **89.00** | 98.52 | 84.70 | 69.47 | 91.93 |
| PC-CLIP | 90.67 | 88.42 | **89.00** | **98.55** | **84.73** | **70.4** | **92.0** |

## C.7 Results with Larger CLIP Model Scales

To evaluate the performance of a larger CLIP model, we trained a ViT-H/14 (Huge) model and compared the results of standard CLIP features with PC-CLIP features. We observe in Table 17 that PC-CLIP still improves over a majority of downstream zeroshot tasks even at larger CLIP model scales, showing that our approach is still effective as we scale up the training data and model size.

# D   Text-to-Image Generation Experiments

## D.1   Text-to-Image Generation Experiment Details

To quantitatively capture an improvement in the ability of PC-CLIP's ability to perform arithmetic in its embedding space, we consider a task of generating photos of images from CIFAR100 (i.e., starting from prompts of "*This is a photo of {class_name}*" where we consider generating for each of the CIFAR100 classes), where we want to assess the ability to add embeddings of specific attributes (e.g., those taken from AwA2 that involve color). Thus, we feed the resulting sum of text embeddings into Stable Diffusion XL and asses how well aligned the generated image corresponds to a text describing the composition of class name and attribute (e.g., "*This is a photo of a blue {class_name}*"). We report the CLIP-Score (Hessel et al., 2021) from a larger CLIP model, namely a ViT-G/14 that has been trained on the LAION-2B English subset (Schuhmann et al., 2022). As a consequence, our CLIP-Scores are computed over 800 different image generations.

Table 17: Performance comparison using ViT-H/14 (Huge) model. Metrics are reported for CIFAR-100, CUB, EuroSAT, Flowers, and SUN datasets.

| Model | CIFAR-100 | CUB | EuroSAT | Flowers | SUN |
|---|---|---|---|---|---|
| CLIP (ViT-H) | 87.60 | 86.42 | 53.56 | **89.06** | 75.64 |
| PC-CLIP (ViT-H) | **87.85** | **86.85** | **54.48** | 88.91 | **75.96** |

## D.2  Text-to Image Visualization Details

To qualitatively evaluate the alignment of our learned embedding space, we can inspect resulting image generations. For instance, we can start with the text embedding corresponding to the prompt of "*A photo of the forest*" and add the text embedding a comparative-based description, such as "*Much more denser forest with lots of trees and a snowier background...*"; the resulting embedding should capture these subtler notions without losing much information from the original prompt. In some cases, as highlighted in our generations, there is improved visual quality when using PC-CLIP as our first text encoder, particularly for more fine-grained features in the generated images.

## D.3  Additional Text-to-Image Generations

We now present additional text-to-image visualizations of our embedding space in Figure 6 and 7. We generally observe that image generations are similar between CLIP + Stable Diffusion (using the model as is) and PC-CLIP + Stable Diffusion (when we swap out the text encoder with our finetuned model) produce very similar image generations, with the caveat that our generations slightly better in maintaining visual coherence and consistency with text descriptions. Aside from the generations given in the main text, we observe in Figure 6 that CLIP + Stable Diffusion is unable to reflect the information in the concept of "*snowier*", given that the forest bushes are less white.

However, we do remark that CLIP + Stable Diffusion is already able to capture the notion of barren trees without leaves, as our model does as well. Similarly, when we perform this semantic arithmetic, it leads to the degradation of visual quality in the cat generation, as depicted by the visual artifact in the cat's tail. However, the overall generation otherwise looks similar and captures the notion of the color orange. We also remark that there are other cases, when performance is roughly the same (see Figure 7). Overall, we remark that our model's embedding space can reflect arithmetic without much loss in the overall quality of the textual features. Further improving the compositional generalization ability of diffusion models is a separate question and an interesting area of future research.

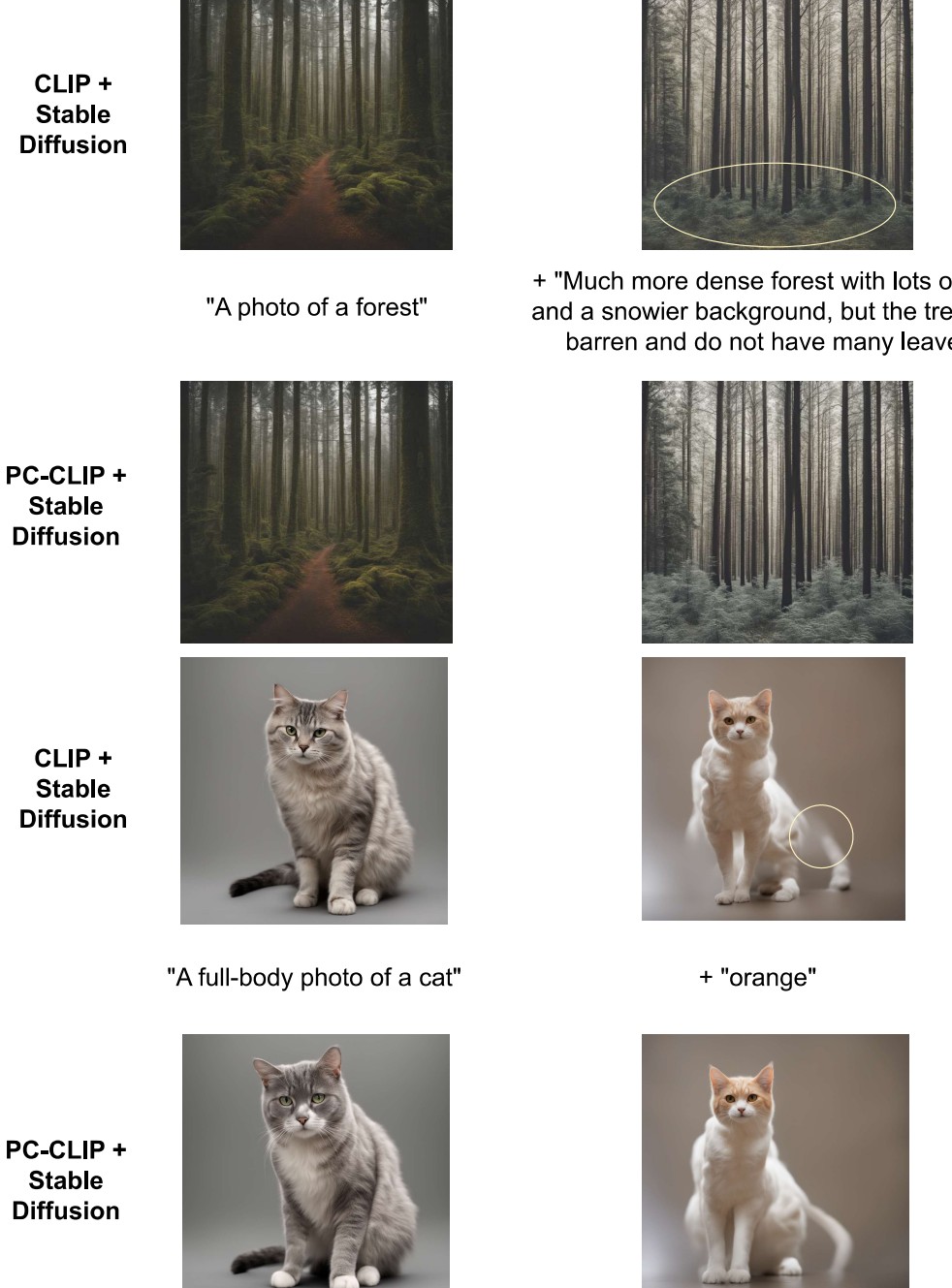

Figure 6: Additional generations from CLIP + Stable Diffusion and PC-CLIP + Stable Diffusion, with relatively shorter descriptions of comparison-based prompts added to the original prompt. Yellow circles capture issues with generations; in the first image, the bushes do not represent any notion of snow, and in the second image, the cat's tail is incomplete.

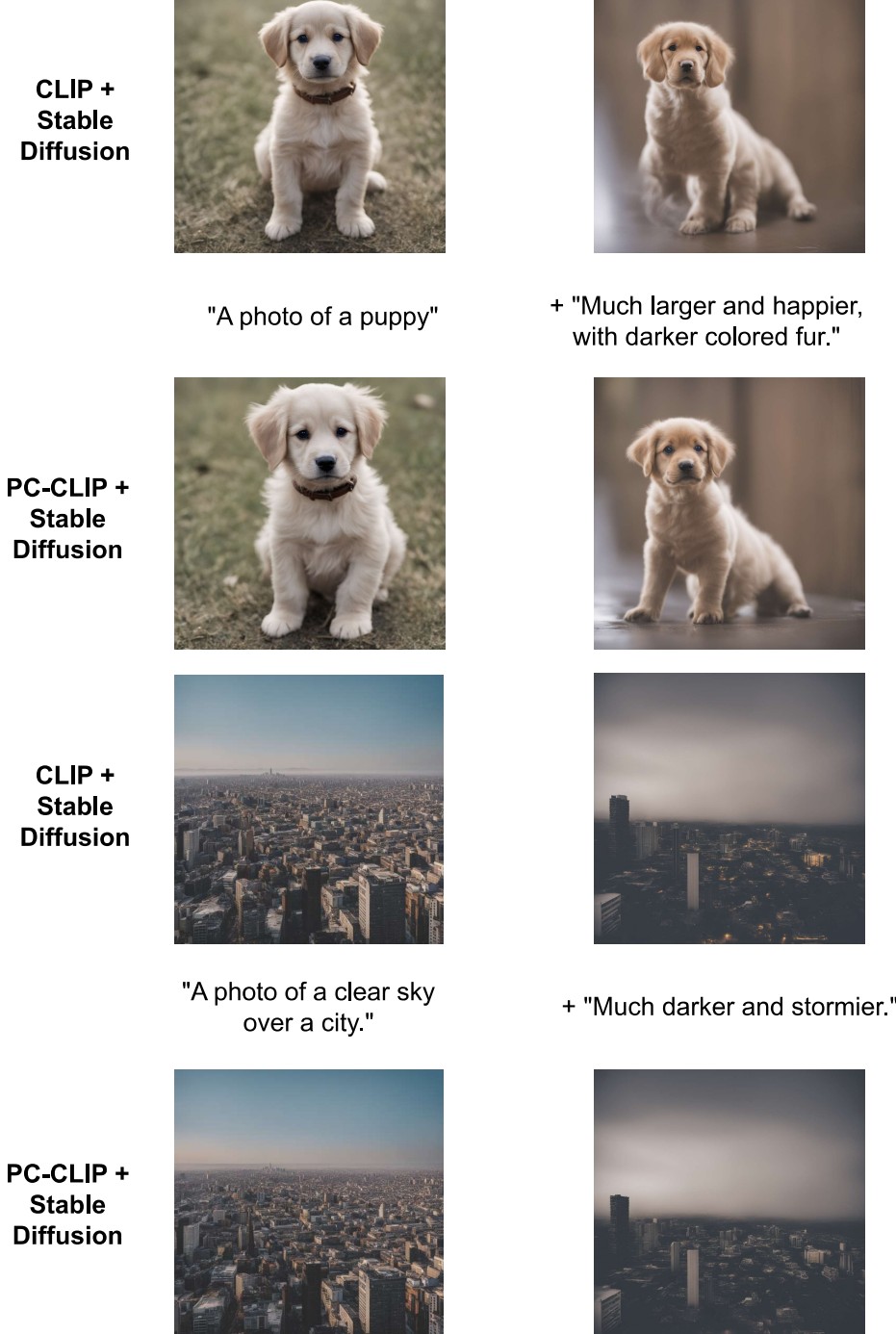

Figure 7: Even more additional generations from CLIP + Stable Diffusion and PC-CLIP + Stable Diffusion, with relatively shorter descriptions of comparison-based prompts added to the original prompt.

