# OpenReview forum: "Finetuning CLIP to Reason about Pairwise Differences"
_TMLR — Accepted by TMLR_

### Review · Reviewer_cisf · 2025-03-10

**Summary Of Contributions:**

This paper introduces PC-CLIP, a variant of CLIP fine-tuned with a Pairwise difference-based loss using synthetic data. This approach is motivated by the inability of the CLIP embedding space to allow for analogies using vector arithmetic.

To construct the synthetic data an LLM is used to, given two captions, return a textual description of the difference between the images. Subsequently, a CLIP model is fine-tuned such that the difference between the image embeddings is similar to the embedding of the difference synthetic difference description.

PC-CLIP is evaluated on a variety of classification tasks, including zero-shot and a proposed task of difference-based classification. Fairly consistent gains are shown for zero-shot classification, with more notable differences for difference-based classification. Additionally, a text-to-image generation experiment is performed.

**Audience:**

Yes

**Broader Impact Concerns:**

Paper contains no broader impact statement and I do not have any concerns that would require adding one.

**Claims And Evidence:**

No

**Requested Changes:**

**Changes related to weaknesses**:
- Incorporate sufficient details in main body of paper to understand experiments without needing the appendix.
- (Critical) Clarity the experiments done in 4.4 - this section lacks details to understand the experiments, particularly the work that led to the results in Table 6.
- (Critical) Evaluate PC-CLIP on cross-modal retrieval
- (Critical) Evaluate and give insight into the quality of the generated synthetic data
- I believe the current image generation experiments do not benefit the paper, they could either be changed or removed.

**Minor changes**:
- It seems inaccurate to state in the introduction that CLIP-like VLM can "These models can also generate images corresponding to user-specified text prompts" as without additional models and training they cannot.
- In 3.1: "Appendix Appendix C.4"
- In 4.3: "(see the bottom two rows in Table 2)" should this refer to the bottom four rows?
- I question the usage of the word 'reason' in the title of the paper

**Strengths And Weaknesses:**

**Strengths**:

The paper rightfully and clearly identifies a problem with CLIP (and contrastively trained VLM) in that the embedding space for such models seems to lack certain properties. The idea to overcome this issue by training on embeddings of differences is interesting and very promising, and using an LLM to obtain synthetic descriptions of these differences seems like a reasonable choice.

**Weaknesses**:
- A number of sections (4.4 in particular) are hard to understand without consulting the Appendix. The clarity of the paper would improve by moving some of the explanations from the appendix to the main paper.
- Mismatch premise and implementation. The problem of the paper is introduced starting from vector arithmetic, but then the problem is recast to pairwise differences without ever returning to vector arithmetic. The chosen implementation of pairwise differences is also rather particular, because the abstract states "We finetune CLIP so that the differences in image embedding space correspond to text
descriptions of the image differences" but since the image encoder is frozen, it is actually more the reverse - the embedding of a description of differences is aligned with a difference between two image embeddings. However, this seems to then presume that the difference in image embeddings is already meaningful based on the pretrained CLIP, and rather the challenge is to align this to statements about differences.
   - Is it correct that only the text encoder is fine-tuned? If so, a clearer explanation of the relationship between the current approach and vector arithmetic/pairwise differences is necessary, as then it would appear no new capabilities are obtained with relation to these, but rather the main gain is due to more exposure to textual statements that discuss differences.
   - This then also brings into question the difference-based classification task, which seems rather specific to this training setup
- No results are reported on retrieval tasks - image2text and text2image are two primary tasks for evaluating CLIP, it would be important to have results on these to understand how this new approach influences these metrics.
- Text-to-image generation experiment is in itself unconvincing;
    - In part this is due to the details given in Appendix D, where it is indicated that this actually concerns a different CLIP variant than the rest of the paper as well as only 1 out of 2 Stable Diffusion XL text encoders.
    - The other part relates to the task itself and the results. The setup of the task is somewhat hard to understand, as it confounds different aspects (e.g., the ability to embedding arithmetic and the ability to accurately encode the text). Moreover, the results mainly seem to indicate that the CLIP version results in artifacts in the generation result (as highlighted by the circled results), which does relate to the embedding arithmetic but it does not support whether PC-CLIP does the arithmetic in a more semantically meaningful way. The difference in CLIPscore is also hard to judge without an additional baseline (i.e., is a 0.01 difference a lot for this task?)
- While in the discussion section it is acknowledged that the method to generate the synthetic difference is based on imperfect information and that LLMs are prone to hallucination it seems insufficient to leave this there. As described in appendix B.2., a relatively simple filtering procedure already discards almost half the generations, and this procedure does not consider the semantics of the generations. More insight into the synthetic data and how the quality of this data relates to the quality of the learned embeddings seems necessary.

---

> ### Author Response · Authors · 2025-05-04
> **Author Response**
>
> Thank you for your detailed feedback and suggestions! We appreciate the thoughtful comments and feel that they have helped us improve the quality and presentation of our paper. We now address individual comments below:
>
> > **Incorporate sufficient details in main body of paper to understand experiments without needing the appendix.**
>
> Thank you for your feedback — we have incorporated your suggestions into our main text to make the experimental settings clearer. Please see our changes in red in the updated revision.
>
> > **(Critical) Clarify the experiments done in 4.4 - this section lacks details to understand the experiments, particularly the work that led to the results in Table 6.**
>
> Thanks for your feedback! We have significantly expanded the description of the experiments corresponding to Table 6 within the main paper. This includes clarifying how difference text prompts were generated, how distances between class and difference embeddings were computed, and how we interpret the results in terms of the localization of relational differences in the embedding space.
>
> > **(Critical) Evaluate PC-CLIP on cross-modal retrieval**
>
> Thank you for this suggestion! We have added a new cross-modal retrieval evaluation on the Flickr30K dataset, now reported in Table 6 of the paper. We also replicate the results here for clarity:
> | Model   | Image-to-Text R@1 | Image-to-Text R@5 | Image-to-Text R@10 | Text-to-Image R@1 | Text-to-Image R@5 | Text-to-Image R@10 | Mean Recall |
> |-|---|--|--|--|--|-|--|
> | CLIP    | 85.30% | 97.20% | 99.20% | 64.88%   |  87.28%  | 92.02% | 87.65%  |
> | PC-CLIP | 84.80% | 97.70% | 99.20% | 69.80% | 89.86% | 94.28% | 89.27% |
>
> We observe that PC-CLIP improves performance for text-to-image retrieval while keeping performance roughly the same on image-to-text retrieval. We also present mean recall (e.g., averaging over both text-to-image and image-to-text and over all different top-k values). We find that using PC-CLIP in retrieval improves overall mean recall.
>
> > **(Critical) Evaluate and give insight into the quality of the generated synthetic data**
>
> Thank you for this suggestion! In Section 4.6 of the revised paper, we now provide a detailed discussion of the quality of synthetic generations. Specifically, we analyze 300 examples of synthetic text-image difference pairs using GPT-4o as an automatic judge. The distribution of scores (shown in Figure 5) indicates that a large fraction of instances are of good quality (mean score = 0.75), but imperfections remain.
>
> Through manual inspection, we find that lower-quality generations typically occur when important visual features are absent from the original COCO captions (e.g., no mention of details on a traffic sign or background scenery), leading to incomplete synthetic differences.
>
> Despite these imperfections, we find that the synthetic data provides a strong training signal at scale, enabling meaningful improvements across a range of tasks. Furthermore, as we show in Appendix C.4, our finetuning procedure is robust to noise in the generated differences: using the full, unfiltered set of synthetic examples still improves downstream performance, demonstrating the method's resilience to imperfect supervision.
>
> > **I believe the current image generation experiments do not benefit the paper, they could either be changed or removed.**
>
> Thank you for this feedback. We have revised this section to be more clear and reflect some experimental changes.
>
> First, we agree that additional experimental details that are necessary, and we have added to our revision. Namely, we now explicitly state that the Stable Diffusion pipeline uses a larger CLIP variant (XL) for generation, which is why we modify this version during these experiments only. This was previously unclear, and we have corrected it in the revision.
>
> We also would like to reemphasize that the main goal of our text-to-image generation experiments is to demonstrate that the PC-CLIP encoder can better reflect semantic operations in embedding space, and we believe that improved generation quality (without visual artifacts in the resulting generations) does reflect this desired takeaway. We believe that the visual artifacts arise in standard CLIP embeddings due to its inability to accurately capture these arithmetic operations.
>
> Finally, we have also improved our evaluation metric to using a large multimodal model (GPT-4o-mini) as a judge instead of the CLIPScore to evaluate the quality of generated images. This judge rates images on a scale of 1–5 for alignment and generation quality. We find that PC-CLIP generations achieve higher mean scores compared to CLIP, confirming the improved semantic alignment in its embeddings (in **Table 5**).
>
> We believe that the changes we have made in our response improve the clarity and completeness of the paper. We are happy to address any further questions or suggestions.

---

> > ### Comment · Reviewer_cisf · 2025-05-08
> >
> > Thank you for the additional information and results. Can you clarify which model was used to obtain the cross-modal CLIP results? The baseline scores seem to be several points below common CLIP results.
> >
> > > Finally, we have also improved our evaluation metric to using a large multimodal model (GPT-4o-mini) as a judge instead of the CLIPScore to evaluate the quality of generated images.
> >
> > Given that there is no clear frame of reference of how good GPT-4o-mini is at evaluating this task, I remain unconvinced about the merits of the generation task.
> >
> > I furthermore also remain unconvinced about my earlier points in relation to the mismatch between the premise and the implementation.

---

> > > ### Author Response · Authors · 2025-05-08
> > > **Response from Authors**
> > >
> > > Thank for your comments!
> > >
> > > > **which model was used to obtain the cross-modal CLIP results**
> > >
> > > We used the OpenCLIP implementation with the
> > > "datacomp_xl_s13b_b90k” checkpoint from the DataComp benchmark [1].
> > >
> > > > **Given that there is no clear frame of reference of how good GPT-4o-mini is at evaluating this task, I remain unconvinced about the merits of the generation task.**
> > >
> > > We agree that evaluating generations with automated metrics poses challenges. However, it is common practice in recent literature to use strong multimodal models as proxy judges. GPT-4o-mini achieves a strong performance of 59.4% on MMMU [2] (see https://openai.com/index/gpt-4o-mini-advancing-cost-efficient-intelligence), which focuses on multimodal reasoning and includes visual question answering tasks. This suggests GPT-4o-mini has reasonably strong capabilities aligned with our evaluation needs.
> > >
> > > > **I furthermore also remain unconvinced about my earlier points in relation to the mismatch between the premise and the implementation.**
> > >
> > > We appreciate the reviewers concern about the premise and implementation. While our submission does assume a fixed image encoder space, incorporating this information into the text encoder does indeed endow PC-CLIP with new capabilities in the text embedding space.
> > >
> > > These improvements manifest not only in the difference-based classification task, but also lead to gains in retrieval and compositional image generation. Moreover, we observe that performance improvements in difference-based classification correlate with gains in standard zero-shot classification and retrieval. This indicates that our fine-tuning strategy does not merely optimize for a contrived objective, but rather enhances the overall semantic structure of the embedding space.
> > >
> > > [1] Gadre, et. al. DataComp: In search of the next generation of multimodal datasets.
> > > [2] Yue, et. al. MMMU: A Massive Multi-discipline Multimodal Understanding and Reasoning Benchmark for Expert AGI.

---

### Review · Reviewer_kn6V · 2025-03-11

**Summary Of Contributions:**

The paper presents Pairwise Comparison CLIP (PC-CLIP), a fine-tuning method designed to enhance CLIP's capability to reason about differences between images in the embedding space. The proposed approach aligns differences in CLIP’s image embeddings with semantically meaningful textual descriptions generated using LLMs. This fine-tuning improves CLIP’s ability to perform comparative reasoning. Additionally, the method enhances CLIP’s performance in attribute-based image ranking and improves zero-shot classification across multiple benchmark datasets. Furthermore, PC-CLIP strengthens the geometric properties of CLIP embeddings, facilitating more structured reasoning, particularly in text-to-image generation through arithmetic operations in the embedding space.

**Audience:**

Yes

**Claims And Evidence:**

Yes

**Requested Changes:**

- Provide an analysis of failure cases where LLM-generated text does not correctly describe the difference between images.

- Examine the conditions under which comparative prompting improves or degrades performance and provide insights into its limitations and possible refinements.

- Visualize the visual representations to illustrate how well differences are localized in the learned embedding space.

**Strengths And Weaknesses:**

**Strengths:**

- The proposed approach effectively enhances CLIP’s capability for comparative reasoning.

- The paper introduces comparative prompting, a novel inference strategy that leverages textual descriptions of class-level differences to improve classification performance.

- The experimental results demonstrate significant improvements in both zero-shot classification and difference-based ranking tasks, validating the effectiveness of the proposed fine-tuning method.

**Weaknesses:**

- The approach relies on LLMs to generate synthetic comparative descriptions from image captions, which may introduce hallucinations or errors.

- Since these descriptions are not explicitly grounded in the images, they may not always be factually accurate or optimal for learning comparative reasoning.

- The fine-tuning process requires computing differences between all image pairs, which scales quadraticallywith dataset size.

---

> ### Author Response · Authors · 2025-05-04
> **Author Response**
>
> Thank you for your detailed feedback and suggestions! We appreciate the thoughtful comments and feel that they have helped us improve our paper. We now address individual comments below:
>
> > **Provide an analysis of failure cases where LLM-generated text does not correctly describe the difference between images.**
>
> Thank you for this suggestion! In Section 4.6 of the revised paper, we now provide a detailed discussion of the quality of synthetic generations. Specifically, we analyze 300 examples of synthetic text-image difference pairs using GPT-4o as an automatic judge. The distribution of scores (shown in Figure 5) indicates that a large fraction of instances are of good quality (mean score = 0.75), but imperfections remain.
>
> Through manual inspection, we find that lower-quality generations typically occur when important visual features are absent from the original COCO captions (e.g., no mention of details on a traffic sign or background scenery), leading to incomplete synthetic differences.
>
> Despite these imperfections, we find that the synthetic data provides a strong training signal at scale, enabling meaningful improvements across a range of tasks. Furthermore, as we show in Appendix C.4, our finetuning procedure is robust to noise in the generated differences: using the full, unfiltered set of synthetic examples still improves downstream performance, demonstrating the method's resilience to imperfect supervision.
>
> > **Examine the conditions under which comparative prompting improves or degrades performance and provide insights into its limitations and possible refinements.**
>
> We appreciate your suggestion and have conducted additional analysis in the revision. We observe that the effectiveness of comparative prompting is strongly tied to the confidence of the standard zeroshot classifier. Specifically, we compute the margin between the top two class probabilities as a measure of classifier confidence. Across the five datasets considered in the paper, we find a strong negative correlation (correlation coefficient = -0.883) between classifier margin and the improvement from comparative prompting.
>
> This result suggests that comparative prompting is most helpful when the model is uncertain about its prediction — intuitively, cases where standard class text embeddings are more ambiguous. As a refinement, we experimented with a simple thresholding strategy: applying comparative prompting only when the model's margin is below 0.1. We find that this selective application further improves performance across all datasets compared to applying comparative prompting uniformly. The corresponding results are now reported in Appendix C.5.
>
> > **Visualize the visual representations to illustrate how well differences are localized in the learned embedding space.**
>
> Thank you for raising this point. As we mention in the paper, our finetuning procedure modifies only the text encoder of CLIP. To visualize how well differences are captured in the text embedding space, we include two key analyses:
> * Quantitative Embedding Distance Analysis (Table 6): We measure how closely LLM-generated text differences align with corresponding class embeddings after finetuning. PC-CLIP shows a substantially smaller gap compared to standard CLIP, indicating that difference representations are better aligned in the text space.
> * Text-to-Image Generation (Figure 4): We use text-to-image generation as a qualitative tool to visualize how PC-CLIP’s text embeddings capture relational concepts. Generations based on PC-CLIP embeddings better incorporate both base class concepts and their corresponding differences, validating the improvements visually.
>
> Together, these results show that the text embedding space learned by PC-CLIP better captures and localizes semantic differences, resulting in improved downstream retrieval, classification, and generation quality.

---

### Review · Reviewer_4d5c · 2025-04-23

**Summary Of Contributions:**

This paper proposes to fine-tune CLIP via differential prompts, which describe the differences between two objects.
The authors show that the resulting PC-CLIP model is compatible with differential-based classification.
Furthermore, PC-CLIP slightly improves conventional zero-shot classification and text-to-image generation.

**Audience:**

Yes

**Claims And Evidence:**

No

**Requested Changes:**

See above. I would appreciate if the authors could address the above weaknesses.

**Strengths And Weaknesses:**

### Strengths and Weaknesses
Overall, the research direction makes sense and is of interest to the research community. However, this paper would benefit from more thorough experimental results to strengthen its arguments.

- **Table 1**. The authors show that PC-CLIP outperforms original CLIP in difference-based classification. This makes sense and shows that the proposed method indeed lets CLIP learn language differentials. However, I am not entirely convinced about when we should consider difference-based classification instead of standard zero-shot classification, especially since the difference-based approach is restricted to binary classification if I understood it correctly.
- **Table 2**. In standard zero-shot classification, the amount of improvement of PC-CLIP over the original CLIP is tiny across multiple datasets. EuroSAT sees a relatively larger improvement (a 2-point bump in accuracy), but the change is still quite insignificant, and the baseline is quite low (55%) so it could be easy to improve upon.
- **Table 3** shows that comparative prompting improves PC-CLIP but not CLIP. This result makes sense, but doesn't sufficiently demonstrate the advantages of PC-CLIP -- even with comparative prompting, PC-CLIP only outperforms CLIP (with or without comparative prompting) on two datasets out of five.
- **Table 4** shows when using LLM-extended class prompts for zero-shot image classification, PC-CLIP noticeably outperforms CLIP only on SUN-397. On the other datasets, PC-CLIP is either similar to CLIP or worse.
- **Table 5**. To me, the text-to-image experiments are the most interesting, as they provide perfect opportunities to showcase PC-CLIP's improved embedding arithmetics. Unfortunately, Table 5 shows that the difference between CLIP and PC-CLIP is tiny, which is then confirmed with the examples in Figure 4. Additional examples in the appendix show larger differences but overall the change is still quite small. Perhaps the authors can consider fine-tuning the text-to-image model with PC-CLIP embeddings?

### Other questions and comments:
- The authors generate comparative prompts with language-only models without visual information. The authors also mention that "this can likely be improved by the advent and usage of large multimodal models that exhibit both image and language understanding." Therefore, I am curious whether this approach has been tried. Can current vision-language models produce better comparative prompts?
- When using PC-CLIP as the text encoder for text-to-image experiments, is the CLIP score computed with PC-CLIP or the original CLIP?
- The authors only update CLIP's text encoder. Does simultaneously updating the text encoder and the audio encoder produce better results?
- The learning rates used to produce PC-CLIP are tiny ($10^{-8}$) and the training seems short (20 epochs over pairs formed by 1000 examples). These could possibly explain the small performance gain over the original CLIP, but why are such tiny learning rates selected? If we diversify the training dataset and mix in standard non-comparative prompts, can we expect better performance?
- Do you think the proposed approach will work when training PC-CLIP from scratch, instead of fine-tuning (not requesting the authors to train from scratch and just hoped to ask for opinions).
- The authors mention that "finding a metric to evaluate text-image alignment is an open research question". One way to reliably evaluate this alignment is through human evaluation. If the authors can show that over numerous CLIP-vs-PC-CLIP example pairs evaluated by multiple human raters, generations of PC-CLIP are significantly more aligned, then the argument "the generated images better match the sum of two text prompts when using PC-CLIP" can become vastly more convincing.

### Typos:
**Section 3.1**: Appendix Appendix C.4.

Finally, I would like to note that I have not personally worked on training multi-modal models, and may not be familiar with some literature in that field. Hence, my review is based on the information presented in this paper.

---

> ### Author Response · Authors · 2025-05-04
> **Author Response (part 1)**
>
> Thank you for your detailed review and your insightful comments! We answer individual comments below:
>
> > **I am not entirely convinced about when we should consider difference-based classification instead of standard zero-shot classification, especially since the difference-based approach is restricted to binary classification if I understood it correctly.**
>
> Our difference-based classification task, though defined over binary pairs, is useful for ranking and retrieval settings where understanding relative attributes is essential (e.g., size, color). It generalizes beyond fixed-class classification by enabling relational reasoning between instances rather than assigning them to discrete categories. We view standard zero-shot classification and difference-based classification as complementary capabilities: while the former maps to classes, the latter captures semantic relationships. Importantly, PC-CLIP achieves stronger performance across both tasks compared to standard CLIP.
>
> > **In standard zero-shot classification, the amount of improvement of PC-CLIP over the original CLIP is tiny across multiple datasets. EuroSAT sees a relatively larger improvement... but the change is still quite insignificant, and the baseline is quite low (55%) so it could be easy to improve upon.**
>
> We note that PC-CLIP achieves consistent improvements across datasets without using any additional labeled data or downstream supervision. This demonstrates that our pairwise difference-based finetuning does not degrade CLIP's original performance—and, in fact, modestly enhances it—even for tasks it was not directly finetuned for. Moreover, while 55% may seem low in absolute terms, we emphasize that zero-shot classification is a challenging setting, and achieving non-trivial gains without downstream supervision is non-trivial. Importantly, our baseline uses the standard approach to defining zeroshot classifiers with CLIP, so they are a reasonable set of baselines.
>
> > **Table 3 shows that comparative prompting improves PC-CLIP but not CLIP. This result makes sense, but doesn't sufficiently demonstrate the advantages of PC-CLIP -- even with comparative prompting, PC-CLIP only outperforms CLIP (with or without comparative prompting) on two datasets out of five.**
>
> We believe that this is a slight misunderstanding from the reviewer. While Table 3 focuses on the relative improvements on highly confused classes, we highlight that across the full datasets (see Table 2), PC-CLIP outperforms CLIP on four out of five tasks after comparative prompting. We have replicated those numbers here:
>
> | **Method** | **CIFAR100** | **CUB**  | **EuroSAT** | **Flowers102** | **SUN397** |
> |-|-|-|-|-|-|
> | CLIP | 85.59 | **81.72**| 54.96 | 81.51 | 72.46 |
> | CLIP + comp | 85.66 | 81.67 | 53.67       | 81.98 | 72.48 |
> | **PC-CLIP + comp**  | **86.08**    | 80.01    | **60.30**   | **82.78**      | **73.64**  |
> The numbers in Table 3 only account for the hardest pairs for each model – not when computed over the full test sets. This table captures relative improvement.
>
> > **Table 4 shows that when using LLM-extended class prompts for zero-shot image classification, PC-CLIP noticeably outperforms CLIP only on SUN-397. On the other datasets, PC-CLIP is either similar to CLIP or worse.**
>
> We would again like to highlight that our results demonstrate that PC-CLIP achieves better performance than CLIP on 3 of the 5 total tasks. This is one of the many results that we show as an added and simultaneous benefit of our PC-CLIP finetuning, which is consistent with our broader finding that difference-based finetuning improves embedding geometry.
>
> > **Perhaps the authors can consider fine-tuning the text-to-image model with PC-CLIP embeddings?**
> Thank you for this suggestion! We appreciate your interest in the text-to-image experiments. In our current work, the use of text-to-image generation is solely intended to visualize the improved properties of the PC-CLIP embeddings. Finetuning text-to-image models such as Stable Diffusion with PC-CLIP embeddings would certainly be an exciting direction, but it falls outside the scope of our paper, which focuses on improving the underlying CLIP embedding space.

---

> > ### Author Response · Authors · 2025-05-04
> > **Author Response (part 2)**
> >
> > We also address your other questions and comments:
> >
> > > **Can current vision-language models produce better comparative prompts?**
> >
> > This is an exciting direction! Recent frontier vision-language models (e.g., GPT-4o) show some ability to reason about relational differences, but systematic evaluations on this capability are still limited. We believe that developing better comparative prompt generation—either through stronger multimodal models or more specialized techniques—is a promising area for future research.
> >
> > > **When using PC-CLIP as the text encoder for text-to-image experiments, is the CLIP score computed with PC-CLIP or the original CLIP?**
> >
> > The CLIP score is computed with the original CLIP, but we note that it is using a larger version of the CLIP model (the XL version), as we generally observe that larger CLIP models have better performance. We also note that we have changed to using a large multimodal model as a judge, which serves as a better evaluation metric.
> >
> > > **The authors only update CLIP's text encoder. Does simultaneously updating the text encoder and the audio encoder produce better results?**
> >
> > Yes, we believe that updating both the text encoder and image encoder would likely produce better results, although given our computational constraints, it is difficult to finetune both simultaneously with large batch sizes (which we note has been observed to be helpful for contrastive learning objectives). We believe that the performance of full finetuning will improve even further with larger batch sizes (e.g., due to more negatives in the contrastive objective).
> >
> > > **The learning rates used to produce PC-CLIP are tiny (10−8) and the training seems short (20 epochs over pairs formed by 1000 examples)... If we diversify the training dataset and mix in standard non-comparative prompts, can we expect better performance?**
> >
> > Yes, we intentionally use small learning rates to preserve information from the original CLIP representations and avoid catastrophic forgetting, as we do not have labeled downstream tasks during finetuning. Incorporating difference-based finetuning into the full pretraining mixture alongside standard contrastive pairs would likely yield even better models, although this would require significant compute. We view this as an exciting avenue for future large-scale training efforts.
> >
> > > **Do you think the proposed approach will work when training PC-CLIP from scratch, instead of fine-tuning (not requesting the authors to train from scratch and just hoped to ask for opinions).**
> >
> > The PC-CLIP objective would still learn relational differences if trained from scratch, but the resulting model would likely struggle on standard tasks (e.g., zeroshot classification), as it may not sufficiently anchor individual concepts without standard contrastive training. Thus, we believe the most effective use of our difference-based objective is in combination with standard CLIP pretraining, either through multi-task objectives or pretraining stage augmentation.
> >
> > > **The authors mention that "finding a metric to evaluate text-image alignment is an open research question". One way to reliably evaluate this alignment is through human evaluation. If the authors can show that over numerous CLIP vs. PC-CLIP example pairs evaluated by multiple human raters, generations of PC-CLIP are significantly more aligned, then the argument "the generated images better match the sum of two text prompts when using PC-CLIP" can become vastly more convincing.**
> >
> > We completely agree that high-quality human evaluation would provide strong evidence. Due to resource constraints, we supplemented our evaluations with GPT-4o-mini as an automated judge. This model scored generations on a scale of 1-5, and we observed a significant improvement when using PC-CLIP embeddings (mean scores of 2.33 vs. 1.75; see Table 5). We believe this provides initial evidence for improved alignment, though we fully acknowledge that rigorous human annotation remains an important direction for future work.

---

> ### Comment · Reviewer_4d5c · 2025-05-15
> **Thank you for the response.**
>
> Thank you for the response! I have a few follow-up questions:
> - Regarding the usefulness of difference-based classification, the authors mention that
> > Our difference-based classification task, though defined over binary pairs, is useful for ranking and retrieval settings where understanding relative attributes is essential (e.g., size, color).
>
> In this case, would it be possible to test PC-CLIP on ranking and retrieval tasks? I am personally not familiar with these tasks, but if open-source datasets for these tasks exist, including those experiments would greatly strengthen this paper.
>
> ***Edit:** I just saw that some retrieval results have been added to the paper (Table 6). Text-to-image retrieval is definitely better now, but image-to-text stays roughly the same. Are there any explanations/intuitions why image-to-text may not improve? Is it because we only do embedding arithmetic for images and not for texts during training?*
>
> - Regarding conventional zero-shot classification results, the authors mention that
> > Across the full datasets (see Table 2), PC-CLIP outperforms CLIP on four out of five tasks after comparative prompting.
>
> While the numbers are indeed higher for PC-CLIP on four out of five tasks, it becomes less convincing if we look at the amount of improvement -- compared with the best CLIP model, PC-CLIP improves CIFAR-100 by 0.42%, Flowers102 by 0.80%, SUN397 by 1.16%, and decreases CUB accuracy by 1.71%. In my opinion, these are not decisive results to support PC-CLIP's advantages. The only clear-cut result is EuroSAT, where PC-CLIP offers an improvement of 5.34%. Overall, I am not fully convinced about PC-CLIP's efficacy on zero-shot classification.
>
> - Fine-tuning text-to-image models -- the authors mention that
> > Finetuning text-to-image models such as Stable Diffusion with PC-CLIP embeddings would certainly be an exciting direction, but it falls outside the scope of our paper, which focuses on improving the underlying CLIP embedding space.
>
> I agree that fine-tuning image generation models per se is not strictly required to support this paper. However, I was not fully convinced by the original text-to-image results, because the changes were tiny. I was proposing fine-tuning the diffusion models to try to provide ideas on how to improve the results.
>
> That said, the GPT-4o-mini results are much more convincing, so I think it's fine not to try fine-tuning at this time.
>
> - The new GPT-4o-mini results are convincing, but I still think it is worth verifying with a small-scale human evaluation if possible. After all, these VLMs are somewhat of a black box.
>
> **Overall, I found the revision highly effective. Good job!**

---

> > ### Author Response · Authors · 2025-05-20
> > **Author Response**
> >
> > Thanks for your continued feedback, and we are happy to hear that you found the revision effective!
> >
> > > **Are there any explanations/intuitions why image-to-text may not improve? Is it because we only do embedding arithmetic for images and not for texts during training?**
> >
> > Yes, we believe that this is a possible explanation for this behavior. As prior work has observed [1], CLIP’s text encoder tends to behave more like a bag-of-words, exhibiting weaker semantic compositionality. Our finetuning helps mitigate this issue by improving the expressivity and structure of the text embeddings, making them better aligned with meaningful differences. This likely has a larger impact in retrieval tasks with text queries, as more discriminative text embeddings yield sharper cosine similarities against a fixed set of image representations.
> >
> > [1] Yuksekgonul, et. al. When and why vision-language models behave like bags-of-words, and what to do about it?
> >
> >  > **Overall, I am not fully convinced about PC-CLIP's efficacy on zero-shot classification.**
> >
> > Thanks for your feedback – while the gains on some tasks are small, we view this as a surprisingly positive result given that our finetuning is entirely task-agnostic and focused on pairwise reasoning. The fact that we observe consistent gains on 4 of 5 tasks (Table 2) despite not optimizing for these objectives suggests that the learned differences generalize in a useful way, even for standard zero-shot settings.
> >
> > > **I still think it is worth verifying with a small-scale human evaluation if possible**
> >
> > We fully agree that human evaluations remain the gold standard, especially for tasks such as the alignment between text prompts and generated images. While we used GPT-4o-mini as a scalable automated judge in our current experiments, we believe that future work could explore the development of human studies to validate and further refine alignment metrics. This is an important direction for the community to explore.

---

### Decision · Action_Editor_SwWh · 2025-06-13

**Recommendation:** Accept with minor revision

**Additional Comments:**

After considering the reviews, rebuttal and paper, AE recommends acceptance of this paper with minor revision. For the minor revision, AE recommends the authors to take another pass over reviewer cisf's comments, in particular the point about "Mismatch premise and implementation". It'd be helpful to make the story more clear and consistent in the introduction and throughout the paper.

**Audience:**

Yes

**Audience Explanation:**

Yes, the finding of difference-based CLIP learning should be of interest to TMLR audience. All reviewers agree with this too.

**Claims And Evidence:**

Yes

**Claims Explanation:**

Yes, the claims of this paper are supported by evidence. All reviewers agree with this too.